# Probing Layer-wise Memorization and Generalization in Deep Neural Networks via Model Stitching

**Aishwarya Gupta, Indranil Saha, Piyush Rai**
*Department of Computer Science and Engineering, IIT Kanpur, India*
*{aishwaryag,isaha,piyush}@cse.iitk.ac.in*

Reviewed on OpenReview: *https://openreview.net/forum?id=XXXX*

## Abstract

It is well-known that deep neural networks can both memorize randomly labeled training data and generalize to unseen inputs. However, despite several prior efforts, the mechanism and dynamics of how and where memorization takes place in the network are still unclear, with contradictory findings in the literature. To address this, we aim to study the functional similarity between the layers of the memorized model and the model that generalizes. Specifically, we leverage model stitching as a tool to enable layer-wise comparison of a memorized (*noisy*) model, trained on a partially noisy-labeled dataset, to that of the generalized (*clean*) model, trained on a clean, noise-free dataset. Our simple but effective approach guides the design of experiments that help shed light on the learning dynamics of different layers in deep neural networks and why models with harmful memorization still generalize well. Our results show that early layers are as important as deeper ones for generalization. We find that "cleaning" the early layers of the noisy model improves the functional similarity of its deeper layers to that of the corresponding layers in the clean model, can drastically reduce memorization and improve generalization. Furthermore, noise fixation up to a certain depth results in generalization similar to that of a noise-free model. However, interestingly, the reverse may not be true. That is, if early layers are noisy but deeper layers are noise-free, then perfect memorization cannot be achieved, emphasizing the dominant role of deeper layers in memorization. Our extensive experiments on four different architectures - customized CNN model, ResNet-18, ResNet-34, and ResNet-50, and three datasets - SVHN, CIFAR-10, and CIFAR-100, with varying levels of noise, consistently corroborate our findings. The code is available at `https://github.com/aishgupta/memorization_stitching.git`.

## 1 Introduction

Deep Neural Nets (DNNs) are widely used across many domains because of their impressive generalization capabilities on unseen inputs. Yet, these models are also prone to memorization, and can fit a randomly labeled dataset perfectly Zhang et al. (2021). A memorized input is known to be inexplicable using generalizable features Wei et al. (2026), and usually represents atypical inputs in long-tailed distributions Feldman & Zhang (2020); Feldman (2020), or the underlying noise/outliers in the training dataset Zhang et al. (2021). Although memorization may occasionally improve downstream performance in imbalanced or long-tailed distributions Maennel et al. (2020); Anagnostidis et al. (2023), it raises significant concerns, including privacy leakage and vulnerability to membership inference attacks Leino & Fredrikson (2020); Carlini et al. (2022). Therefore, understanding when and how memorization occurs is critical for building more robust and interpretable models.

The understanding of the mechanisms and dynamics of memorization in DNNs has intrigued researchers since the seminal work of Zhang et al. Zhang et al. (2021). Numerous diverse approaches have been leveraged to study and find answers to foundational questions, such as how a model can simultaneously memorize noise and generalize to unseen data Krueger et al. (2017); Arpit et al. (2017), whether all layers in a network converge similarly in the presence of label noise , or whether certain layers are more susceptible to memorization than

others Stephenson et al. (2021); Wongso et al. (2023); Maini et al. (2023), and so on. These studies not only helped in understanding how memorization manifests in deep neural networks, but they also laid the foundational work for approaches designed to mitigate memorization. However, despite significant progress, these efforts have sometimes resulted in divergent conclusions, especially regarding the susceptibility and role of the individual layer in memorization. For instance, Stephenson et al. (2021) showed confinement of memorization to deeper layers of the model, while Wongso et al. (2023) showed that early layers are also involved in memorization despite their relatively faster convergence than deeper layers. Furthermore, Maini et al. (2023) localized memorization to a small subset of neurons spread across all layers in the network. All these findings capture different facets of memorization, highlighting the complexity of disentangling memorization from generalization.

In this work, we build on these insights and ask a fundamental question: *What is the functional similarity between the layers of a model that memorize and those that generalize?* Specifically, our goal is to assess the impact of memorization on the functionality of individual layers, which we define in terms of their ability to align with the behavior of their clean counterparts. To answer this, we compare a memorized model, trained on randomly labeled inputs, to a generalized model trained on the clean, noise-free dataset. We leverage the recently proposed stitching layer Csiszárik et al. (2021); Bansal et al. (2021), a linear trainable layer that "stitches" two models by aligning their latent representations, as a tool to enable the comparison of layer-wise functional similarity between models. We stitch the memorized model to the generalized one at varied depths and vice versa, and evaluate all stitched models on two downstream tasks - (i) generalization to unseen test data, and (ii) memorization accuracy on randomly labeled training inputs to quantify similarity/divergence between their generalizing and memorizing functional behavior. Our methodology provides insights into the following fundamental questions:

- How does fixing (replacing) certain layers affect memorization and generalization?

- How does memorization propagate through the network?

- Are all layers equally responsible for memorization, or do some play a more critical role?

- How do the inherent bias of the architecture, underlying noise level, and task complexity influence the impact of memorization at different layers?

Our results uncover consistent and surprising patterns across architectures and datasets, offering new insights into how memorization emerges, evolves, and influences the functional behavior of individual layers. Our contributions and key findings can be summarized as:

- We propose a novel framework for evaluating the impact of memorization on each layer of the DNN model using model stitching, enabling a direct layer-wise functional comparison between models trained with and without noise. To the best of our knowledge, our work is the first of its kind, enabling direct one-on-one functional comparison between memorized and generalized models.

- We experimentally analyze four architectures - CNN, ResNet-18, ResNet-34, and ResNet-50 on three datasets - SVHN, CIFAR-10, and CIFAR-100, with varying extent of noise memorization. Our results consistently demonstrate the following:
    1. We observe that noise memorization impacts each and every layer of the model, and replacing early layers with clean counterparts improves the test-time performance of the model.
    2. If a sufficient number of bottom layers are replaced by their clean counterparts in the noisy model, then often the memorized accuracy of the model drops to a random guess. That is, despite the presence of noisy deep layers, fixing the early layers mitigates memorization. We hypothesize that this is attributed to the feedforward nature of DNNs that compounds noise present in early layers during the forward pass. If the noise in early layers is fixed, noise cascading is also minimized, rendering noise in deeper layers benign.
    3. Deeper layers are necessary for perfect memorization. The insertion of early noisy layers in the clean model shows the emergence of non-trivial memorization, but still, 100% memorization

is only achieved when noise is present in deeper layers as well. This corroborates the initial findings of training/fixing deeper layers to reduce memorization.

4. We observe the influence of the inherent bias of model architecture and task complexity on the extent of memorization in each layer. Specifically, the number of early layers that, if fixed (replaced with clean layers), can mitigate memorization is jointly decided by the task complexity and model architecture.

- We also perform preliminary experiments on more complex and deeper architectures, such as Vision Transformer and ResNet-101, to probe layer-wise memorization and its impact on the model's end-to-end behavior. Our experiments show promising directions that could be explored at length in future works to enhance the understanding of memorization.

## 2 Related Work

**Memorization** DNNs are infamously known to perfectly memorize randomly labeled training inputs when heavily over-parameterized, exhibiting universal finite-sample expressivity Zhang et al. (2017). In the presence of label noise, model training starts with learning simple, generalizable patterns, followed by memorizing specific, randomly labeled training inputs in the later stage Krueger et al. (2017); Arpit et al. (2017). This observation has inspired early-stopping approaches to curb memorization while preserving generalization Liu et al. (2020); Frankle et al. (2020). Moreover, multiple factors such as the inductive bias from the model architecture Zhang et al. (2020), explicit regularization, or training protocols Arpit et al. (2017) play a significant role in how quickly and how much a model can memorize.

Beyond mitigating memorization, considerable work has been done in understanding where and how memorization manifests within a model. A variety of approaches have been proposed to probe memorization, such as prediction depth via replica-based mean-field geometry Stephenson et al. (2021) of each input, manifold analysis techniques Chung et al. (2018), sliced mutual information Wongso et al. (2023), gradient accounting, and layer retraining Maini et al. (2023). Interestingly, each of these methods has captured different facets of memorization, sometimes resulting in divergent conclusions. For instance, Stephenson et al. (2021) analysed the geometric properties of class manifolds throughout training and found that memorization is largely confined to deeper layers. They further showed that rolling back the weights of the final convolution layer to an early epoch can reverse memorization to a great extent. However, Wongso et al. (2023) showed that all layers of the model memorize and early layers stabilize earlier than deeper layers. A recent work Maini et al. (2023) combined techniques like gradient accounting, layer rewinding, and selective retaining to localize memorization within the model. They discovered that memorization is not limited to any single layer; instead, it is confined to a subset of neurons spread across various layers. This suggests that memorization affects all layers of the model (although through a relatively small subset of neurons in each layer), supporting the findings of Wongso et al. (2023).

**Model Stitching** Model stitching is introduced by Lenc & Vedaldi (2015) to study the equivalence of representations. The stitching layer is a learnable linear transformation that connects two models by aligning their intermediate representations. Model stitching has become a popular approach for building new architectures adhering to specific resource and performance requirements by combining components from existing models of varying complexity Yang et al. (2022); Teerapittayanon et al. (2023); Pan et al. (2023; 2024); He et al. (2024); Xu et al. (2024). Additionally, researchers use stitching to probe functional similarity between models' hidden representations, assessing how well cross-wired components perform on the same task Bansal et al. (2021); Csiszárik et al. (2021). Empirical evidence shows that even when two networks' representations score low on metrics like CKA, they can still exhibit high functional similarity, demonstrating that representational and functional similarity capture complementary aspects of how representations relate Bansal et al. (2021). For instance, stitching has revealed strong functional correspondence between adversarially robust and standard (non-robust) models Balogh & Jelasity (2023), despite these networks showing low CKA-based similarity Cianfarani et al. (2022). Accordingly, we employ the stitching layer as a tool to measure functional similarity between a generalized (clean) model and a memorized (noisy) model. Importantly, our setup is more challenging than Balogh & Jelasity (2023) owing to the noisy model trained on partially

randomly corrupted labels, and thus, receiving conflicting supervisory signals compared to the clean model, which is trained exclusively on correctly labeled data.

## 3 Methodology

Our goal is to investigate how memorization manifests across different layers of the model and how it affects generalization. We adopt the experimental setup of Maini et al. (2023), introducing randomly labeled inputs into the training dataset to induce memorization. Next, we train a separate model on clean data without label noise so that none of its training inputs have to be specifically memorized, and use it as a baseline for generalization. The core idea is that the clean model, owing to zero-noise exposure, exhibits ideal generalization behavior and thus, a direct one-on-one comparison of internal representations with the memorized model can uncover the layer-wise functional differences that emerge due to noise memorization.

We follow the definition of functional similarity as defined by Csiszárik et al. (2021); Bansal et al. (2021), and leverage model stitching as a tool to maximally align internal representations between generalized (*clean*) and memorized (*noisy*) models with identical architecture. Here, by "maximization of alignment", we refer to the maximization of the possible functional alignment between representations with respect to the downstream task. A stitched model is constructed by joining two models $f_{r_1}$ and $f_{r_2}$, trained with different noise level $r_1$ and $r_2$ respectively, at some intermediate layer $l$. A trainable linear stitching layer $S$ is inserted to align the output of layer $l$ of $f_{r_1}$ as the input to layer $l+1$ of $f_{r_2}$. The stitching layer $S$ is trained by minimizing the loss function of the stitched model (with weights of $f_{r_1}$ and $f_{r_2}$ frozen), and when the loss converges, the learned stitching layer offers maximum possible functional alignment between the representations.

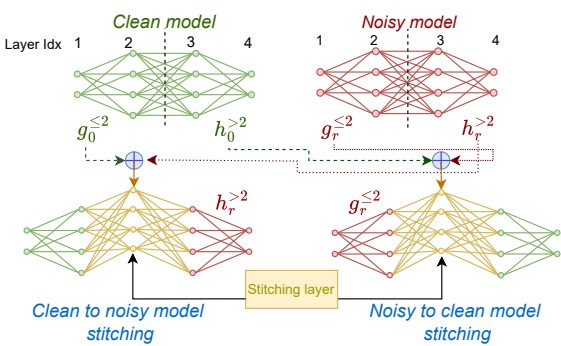

Figure 1: Stitching clean and noisy model at layer idx 2. Clean-to-noisy model stitching consists of a clean base, stitching layer, and a noisy head, while a noisy-to-clean stitched model consists of a noisy base, stitching layer, and a clean head.

This results in a model of the form: $h_{r_2}^{>l}(S(g_{r_1}^{\leq l}))$, where $g_{r_1}^{\leq l}$ and $h_{r_2}^{>l}$ denote the *base* and *head* parts of the stitched model, respectively. Here, we represent the base or early part of the stitched model by $g$, and the head or latter part of the stitched model by $h$. We annotate $g$ with subscript $r_1$ and superscript $\leq l$ to show that part of $f_{r_1}$ up to layer $l$ forms the base of the stitched model. Similarly, $h_{r_2}^{>l}$ shows that part of the model $f_{r_2}$ from layer $(l+1)$ onwards forms the head of the stitched model. From here on, for brevity, "stitching at layer $l$" refers to inserting a trainable stitching layer between the output of layer $l$ in the base model and the input to layer $(l+1)$ in the head model.

We experiment with two stitching directions:

- **Clean-to-noisy** $h_r^{>l}(S(g_0^{\leq l}))$, where the clean model $f_0$ up to layer $l$ forms the base, and the latter part of the noisy model $f_r$ (starting from layer $(l+1)$) forms the head of the stitched model.

- **Noisy-to-clean** $h_0^{>l}(S(g_r^{\leq l}))$, where the noisy base ($f_r$ up to layer $l$) is stitched to clean head ($f_0$ from layer $l+1$ onwards).

The stitching process is visualized in Figure 1. The accuracy of the stitched model reflects how well the intermediate representations of the two models can be linearly transformed and aligned. As a result, the performance gap between the stitched model and its original constituent models serves as a proxy for functional dissimilarity caused by memorization. For example, the performance gap between $h_0^{>l}(S(g_r^{\leq l}))$ and $f_0$ arises from replacing the early layers (up to layer $l$) with the corresponding layers of the noisy model $f_r$. This performance gap w.r.t $f_0$ serves as a measure of the collective functional similarity between the early layers (1 to $l$) of the clean $f_0$ and noisy $f_r$ model. Similarly, performance comparison of $h_r^{>l}(S(g_0^{\leq l}))$ to $f_r$ evaluates the functional similarity between the stitched part of the clean model and the unstitched part of the noisy

model (i.e., $f_0^{1:l}$ and $f_r^{1:l}$). However, the key component is the dataset used to train the stitching layer $S$, on which intermediate representations are maximally aligned between models. In our setup, we train stitching layers $S$ on the correctly-labeled validation dataset $\mathcal{D}^{val}$, unseen by both $f_0$ and $f_r$. We choose the validation set $\mathcal{D}^{val}$ instead of the training set of either the $f_0$ or $f_r$ model so that the training of the stitching layer does not interfere with divergent representations of memorized inputs learned by the two models. Furthermore, training a stitching layer on $\mathcal{D}^{val}$, which is disjoint from training data, ensures alignment of generalizable representations rather than memorized patterns, limiting memorization to only the stitched part of the noisy model.

## 4   Experiment Setup

We understand memorization dynamics in Deep Neural Networks (DNNs) using four different architectures - a customized CNN comprising three convolutions and four fully connected layers, ResNet-18, ResNet-34, and ResNet-50 He et al. (2016). Our experiment spans three datasets - SVHN Netzer et al. (2011), CIFAR-10, and CIFAR-100 Krizhevsky et al. (2009). We train a customized CNN on SVHN, ResNet-18, and ResNet-34 on CIFAR-10, and ResNet-34 and ResNet-50 on CIFAR-100, resulting in a total of 5 distinct models. We also perform preliminary experiments on ResNet-101 and ViT-Small trained on CIFAR-100 and mention experimental findings in Appendix A.2 and A.3, respectively. To study memorization, we introduce label noise into the training data by assigning random labels to a fraction of training images. We consider noise levels $r = \{5\%, 10\%, 15\%, 20\%, 25\%, 37.5\%, 50\%\}$ and train each model on a noisy dataset $\mathcal{D}_r$ until its training accuracy exceeds 99% to ensure noise memorization. Note that each noise-injected training dataset $\mathcal{D}^r$ comprises both correctly labeled inputs $\mathcal{D}_c^r$ and randomly labeled inputs $\mathcal{D}_n^r$. We also train a model on the clean dataset $\mathcal{D}^0$ (i.e., $r = 0$), where all training images are correctly labeled to serve as a generalization baseline.

A model trained on $\mathcal{D}^0$ has been exposed to zero random noise during training and does not memorize any noisy-labeled training input. We refer to this model as the *noise-free* or *clean model* and use it as a generalized model for comparison. Contrary to this, all models trained on noisy datasets must memorize randomly assigned labels during training and thus are referred to as *noisy models*. Notably, while the same images appear in $\mathcal{D}_n^r$ and $\mathcal{D}^0$, their labels differ, inducing divergent learned representations, particularly for $\mathcal{D}_n^r$. By divergent, we refer to the latent representations of the randomly labeled inputs in the clean and noisy models. The clean and noisy model maps them to correct and random labels, respectively, resulting in different latent representations. Such representations are termed "divergent" as one cannot be mapped to another explicitly during training on the validation dataset. Our selection of validation as a training dataset for stitching layers ensures that these divergent representations are not at all explicitly aligned, restricting noise localization only to the noisy part of the stitched model, and thus, playing a crucial role in understanding the manifestation of memorization and the generalization drop in noisy models.

In our experiments, we have performed stitching for the CNN model after every convolution and fully connected layer, and for ResNet models, it is performed after every residual block. To get stitched models with the clean base and noisy head $h_r^{>l}(S(g_0^{\leq l}))$, we start with a noisy model $f_r$ and incrementally replace its initial layers/blocks with those from a clean model. The process begins with stitching the first layer/block of the clean model onto the remaining layers/blocks of the noisy model. Next, we repeat it by incrementally increasing the number of layers/blocks of the clean model to be stitched onto the corresponding remaining layers/blocks of the noisy model. So, in the first round, only the first layer/block is stitched. In the second round, we stitch up to the first two layers/blocks, and so on, each time replacing a longer prefix of the noisy model with the corresponding clean layers/blocks. This process continues until all layers/blocks, except for the last one, of the noisy model are replaced by their counterparts from the clean model. Similarly, we start with a clean model $f_0$ and incrementally replace its initial layers/blocks with those from a noisy model, resulting in a series of stitched models with the noisy base and clean head $h_0^{>l}(S(g_r^{\leq l}))$, each corresponding to a different stitching depth $l$. The training convergence plots of all stitching layers are presented in Appendix A.6.

## 5    Results and Findings

We organize the results into two main parts: the memorization behavior and the generalization performance of stitched models, aiding in the collective understanding of the functional behavior of the clean and noisy models.

### 5.1    Memorization

The stitching layer is trained to maximize the representation alignment between the two models on the correctly labeled validation dataset. As a result, the stitching layer cannot align the divergent representations learned by the clean and noisy model on $D_n^r$ dataset, limiting noise memorization in the stitched model to the constituent part of the noisy model. This raises the following key question: 1) Do stitched models comprising a partial clean model still exhibit memorization when evaluated on $\mathcal{D}_n^r$? 2) If yes, then to what extent? 3) Does the stitching depth impact the extent of memorization in the stitched model? We address these questions by evaluating the accuracy of stitched models on the noisy subset of the training dataset, i.e., $D_r^n$, and use it as a measure to quantify and compare memorization. In all figures, the accuracy of memorized inputs is plotted on the Y-axis as a function of stitch depth $l$ on the X-axis for all stitched models, and each line plot corresponds to a specific value of noise $r$. Note that zero memorization means the stitched model does not recall those specific random label assignments, resulting in a random accuracy ($100/N$, $N$ being the number of classes) on memorized inputs.

### 5.1.1    Clean-to-Noisy Stitching $h_r^{>l}(S(g_0^{\leq l}))$

The accuracy of all noisy models, stitched models $h_r^{>l}(S(g_0^{\leq l}))$, and the clean model on $D_r^n$ is plotted in Figure 2. Each subplot corresponds to a specific architecture and dataset. The key observations are summarized below.

M.1 **On simple tasks, fixing the very first layer/block substantially reduces memorization**. For smaller datasets such as SVHN and CIFAR-10, stitching in just one or two clean layers/blocks dramatically suppresses memorization (quantified as accuracy on randomly labeled training inputs $D_r^n$), regardless of noise ratio $r$. For example, at $r = 50\%$, the memorized accuracy drops below 50% for both datasets by replacing only the first noisy layer/block by its clean counterpart. In contrast, for a more challenging dataset like CIFAR-100, achieving a similar drop at $r = 50\%$ requires stitching clean layers up to at least the middle block.

M.2 **Stitching up to the minimum clean prefix $l_c^m$ reduces memorization to random.** Across architectures and datasets, memorization consistently decreases as more clean bottom layers are stitched in, eventually collapsing to near-random accuracy once the clean base reaches a critical depth $l_c^m$. This $l_c^m$ represents the *minimum clean depth* needed to "forget" memorization. Notably, the stitch depth $l_c^m$ never lies among the final layers, indicating that noisy upper layers alone cannot sustain memorized behavior once early noisy layers are replaced by their clean counterparts.

M.3 **Architecture and task complexity strongly influence $l_c^m$.** The required stitch depth $l_c^m$ is shaped by both architectural inductive biases and dataset difficulty. CNNs exhibit a steep drop in memorization after introducing the first clean layer, while ResNets on CIFAR-10 show a slower decline—likely due to residual connections. On CIFAR-100, memorization initially decreases very gradually, followed by a sharp drop only after a sufficiently large clean prefix is stitched in. Even within the same architecture (e.g., ResNet-34), the pattern of memorization decay differs notably between CIFAR-10 and CIFAR-100. Also, higher noise ratios produce a larger set of memorized examples, making the corresponding model more sensitive to layer replacement. Consequently, stitching clean bottom layers induces a sharper decline in memorization for larger values of $r$. Nevertheless, for a given architecture and model, the decreasing trend of memorization exhibits a similar pattern across all noise ratios.

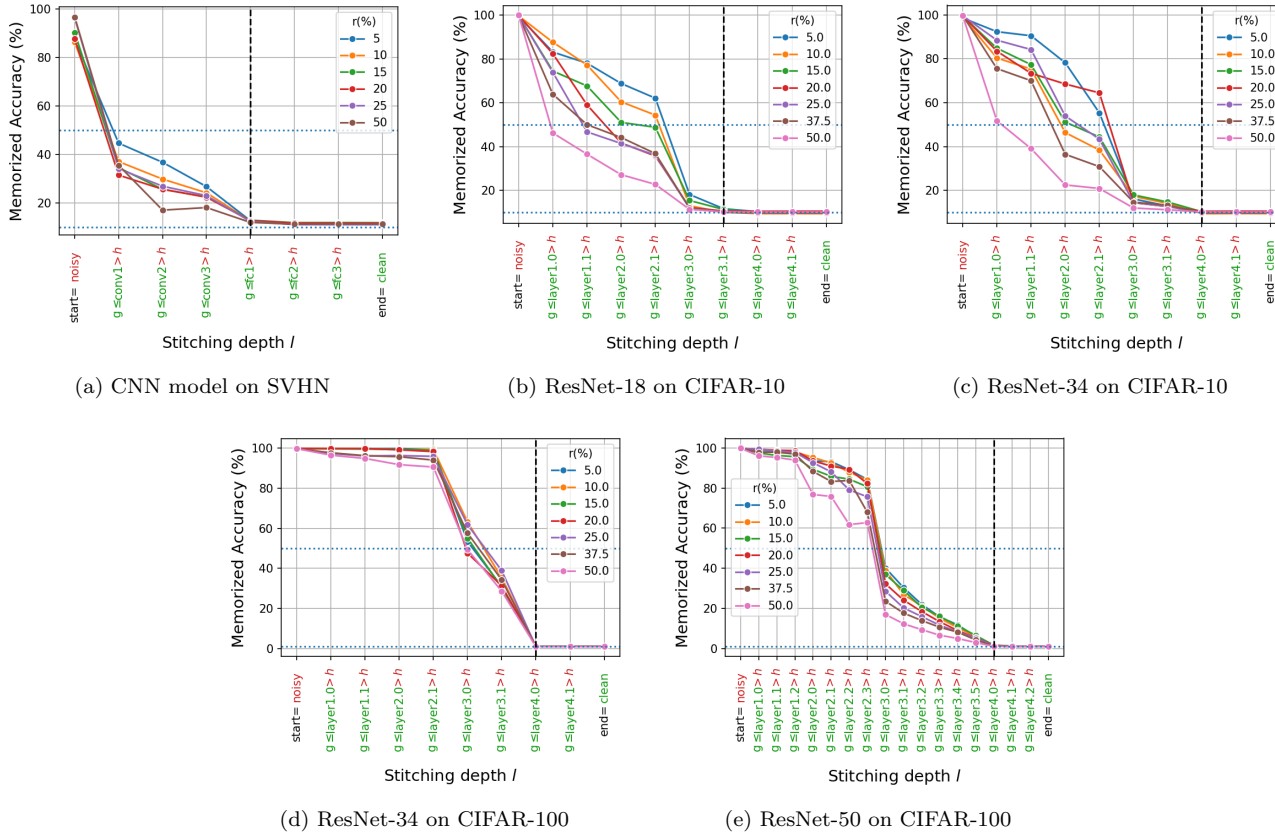

Figure 2: Clean-to-Noisy Stitching $h_r^{>l}(S(g_0^{\leq l}))$: Accuracy of stitched models on randomly labeled training inputs $D_n^r$ of the constituent noisy model. Each xtick corresponds to a stitched model comprising clean base up to the layer $l$, stitching layer, and the noisy head starting from layer $l+1$ to the last layer. The black vertical line shows the minimum depth of clean base $l_c^m$ required to mitigate memorization in noisy models.

### 5.1.2 Noisy-to-Clean Stitching $h_r^{>l}(S(g_0^{\leq l}))$

The memorization accuracy (i.e., accuracy on randomly labeled training inputs) of stitched models $h_r^{>l}(S(g_0^{\leq l}))$ along with the clean and noisy baselines is shown in Figure 3. The key observations are:

M.4 **Early noisy layers do not immediately induce memorization.** While stitching in early clean layers in the noisy model significantly reduces memorization (as seen in clean-to-noisy stitching Figure 2), the converse does not hold. Adding a few noisy early layers to a clean model does not immediately increase memorization. Instead, memorized accuracy remains close to random for shallow stitching depths and begins to rise slowly only when the number of noisy bottom layers exceeds the number of remaining clean upper layers.

M.5 **Memorization emerges only after surpassing the minimum noisy prefix $l_n^m$, but never reaches** $100\%$**.** Once the stitching depth exceeds a certain depth $l_n^m$, the stitched model begins to show nontrivial memorization - even though its head remains noise-free. This depth is referred to as *minimum noisy suffix* needed for the emergence of memorization. However, noisy-to-clean stitched models never show full memorization, even if all layers except the output layer are noisy. For example, when only the output layer is clean, the CNN model trained on SVHN achieves only 20-30% accuracy on random labels, ResNet-34 achieves at most 55% on CIFAR-10 and 30% on CIFAR-100, and ResNet-50 reaches $\approx 60\%$ on CIFAR-100. This demonstrates the dominant role of the clean head in limiting the model's ability to memorize arbitrary labels.

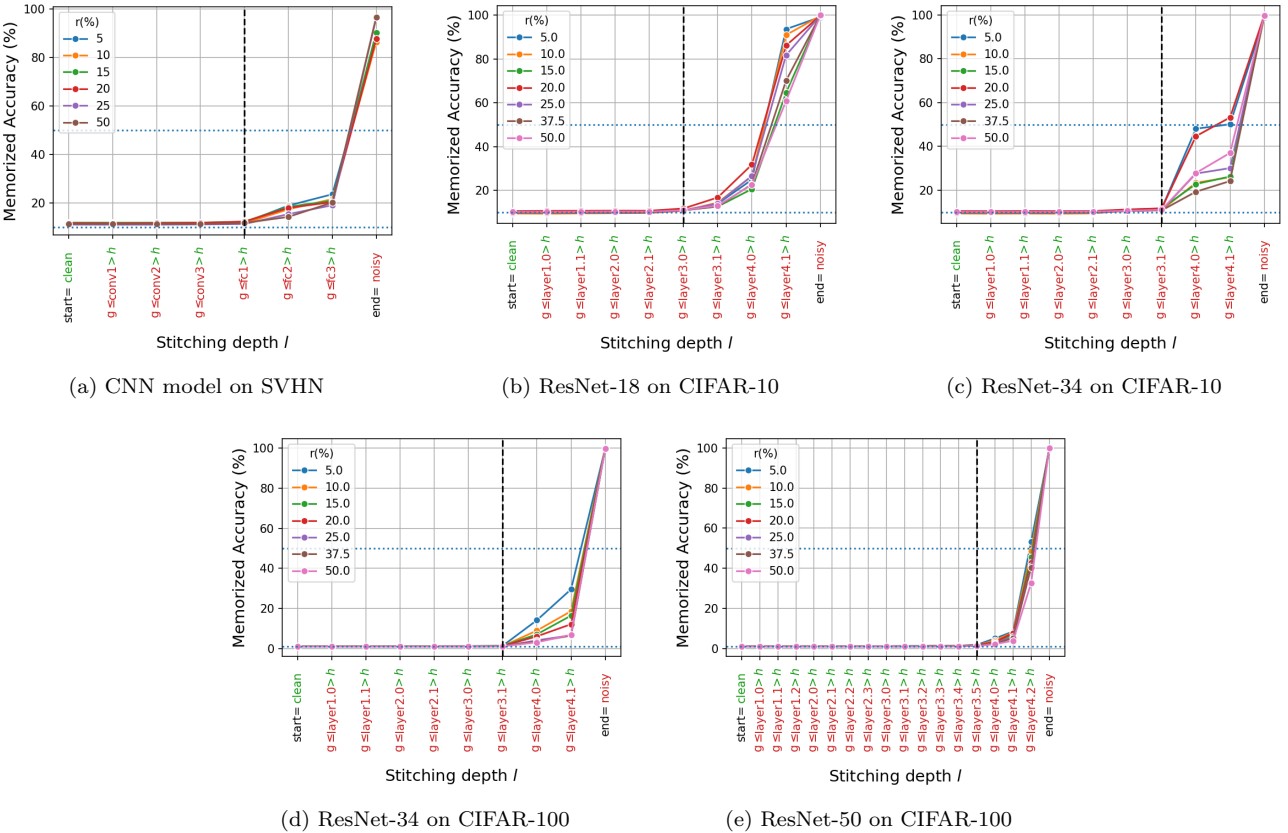

Figure 3: Noisy-to-Clean Stitching $h_0^{>l}(S(g_{\overline{r}}^{\leq l}))$: Accuracy of stitched models on randomly labeled training inputs $D_n^r$. Each xtick corresponds to a stitched model comprising noisy base up to the layer $l$, stitching layer and the clean head starting from layer $l+1$ to the last layer. The black vertical line shows the minimum depth of the noisy base $l_n^m$ required for the emergence of memorization in the clean model.

## 5.2 Generalization

We assess the generalization of stitched models to address the following key questions: 1) Does the noisy model's generalization improve as a deeper clean base is stitched in? 2) Does the clean model's generalization degrade as more noisy layers are stitched in? 3) Can full generalization be maintained even when a part of the model remains noisy? We evaluate and compare the accuracy of stitched models against both the clean and noisy models on the unseen test dataset ($\mathcal{D}^{test}$). Note that when the test accuracy of the stitched model matches that of the clean model, it indicates zero adverse impact of the training noise on its functional behavior, and thus, can be referred to as full generalization.

### 5.2.1 Clean-to-Noisy Stitching $h_r^{>l}(S(g_0^{\leq l}))$

We stitch the early clean layers ($g_0^{\leq l}$) to the latter noisy layers ($h_r^{>l}$) at varied depths, and report their test accuracy in Figure 4. The main observations are summarized below.

G.1 **Replacing even the first noisy layer/block significantly improves generalization.** Across all architectures, stitching in just the first layer/block yields a notable boost in test accuracy, especially for high noise percentages. For instance, at $r = 50\%$, stitching in the first clean convolution layer in the CNN model or the first clean residual block in ResNet-18 boosted test performance by roughly 8-10%. This indicates that the early layers of the noisy model have also learned some non-generalizable features, and swapping them out has an outsized positive effect.

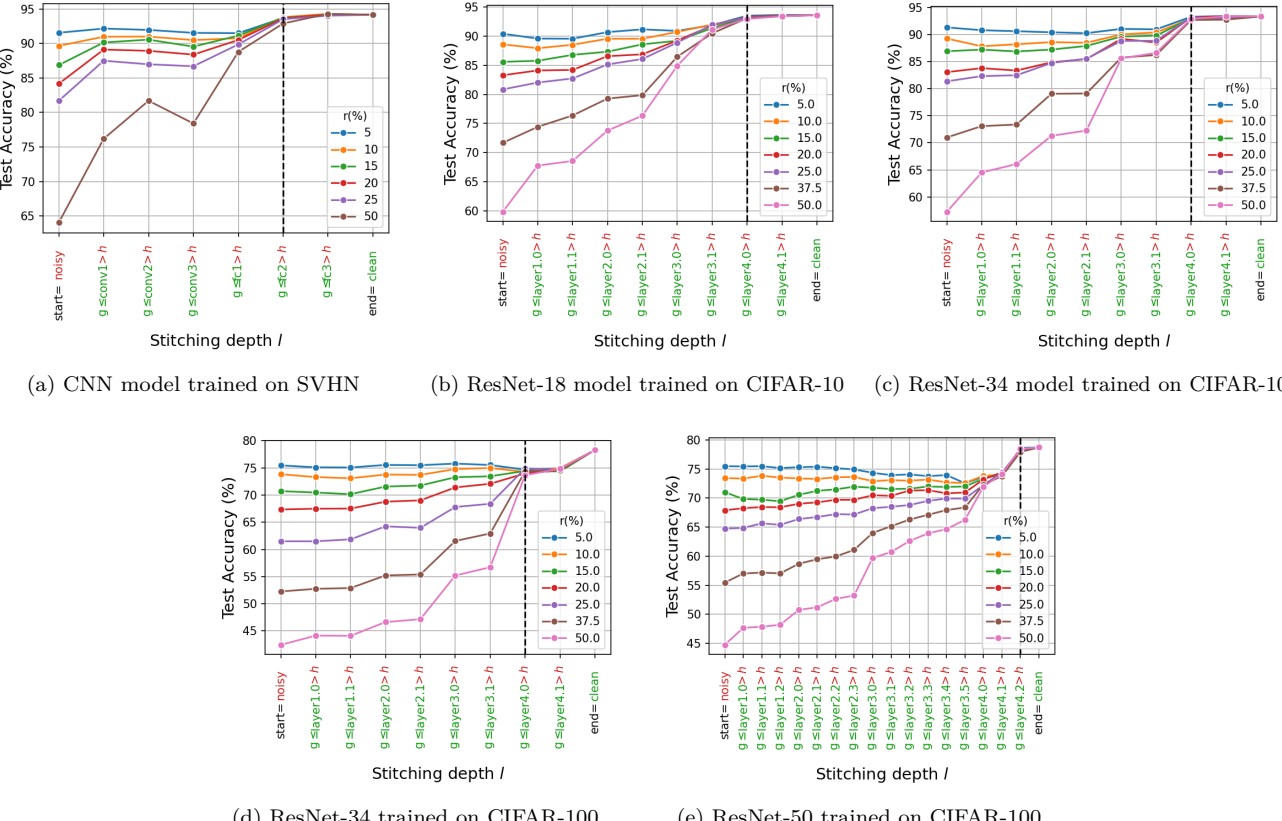

Figure 4: Clean-to-Noisy Stitching $h_r^{>l}(S(g_0^{\leq l}))$: Accuracy of stitched models on unseen test dataset. Each xtick corresponds to a stitched model comprising clean base up to the layer $l$, stitching layer, and the noisy head starting from layer $l+1$ to the last layer. The black vertical line shows the minimum depth of the clean base $l_c^g$ required for restoring (almost) full generalization in the noisy model.

G.2 **Stitching beyond the sufficient clean prefix $l_c^g$ yields negligible gains.** The test performance of the stitched models lies between that of the constituent noisy $f_r$ and clean $f_0$ model. The test accuracy consistently improves with an increase in stitching depth $l$ (more clean layers in the base), eventually matching the performance of the clean model once the clean base reaches a critical depth $l_c^g$. This $l_c^g$ represents the *minimum clean depth needed to restore full generalization.* Interestingly, $l_c^g$ never corresponds to the final layer, showing that despite the noise influence present in the head $h_r^{>l}$ layers of the model, replacing an appropriate subset of early layers is enough to achieve generalization.

G.3 **Architecture and task complexity influence the depth of $l_c^g$.** The location of $l_c^g$ depends on both architectural inductive biases and dataset difficulty. For example, in the CNN model trained on SVHN, $l_c = \texttt{fc2}$ lies near the end of the architecture, whereas in ResNet-18 trained on CIFAR-10, $l_c^g = \texttt{layer4.0}$ occurs much earlier relative to the model's depth. Even within the same architecture, such as ResNet-34, the depth of $l_c^g$ differs across datasets. On CIFAR-10, $l_c = \texttt{layer4.0}$, the second last residual block (similar to ResNet-18 trained on CIFAR-10), but for the more complex CIFAR-100 dataset, performance continues improving up to the last residual block $\texttt{layer4.1}$, analogous to ResNet-50 where $l_c = \texttt{layer4.2}$. Strikingly, for any given architecture-dataset pair, all stitched models converge to similar test accuracy at $l_c^g$, regardless of the noise $r$, and exhibit similar improvement trajectories.

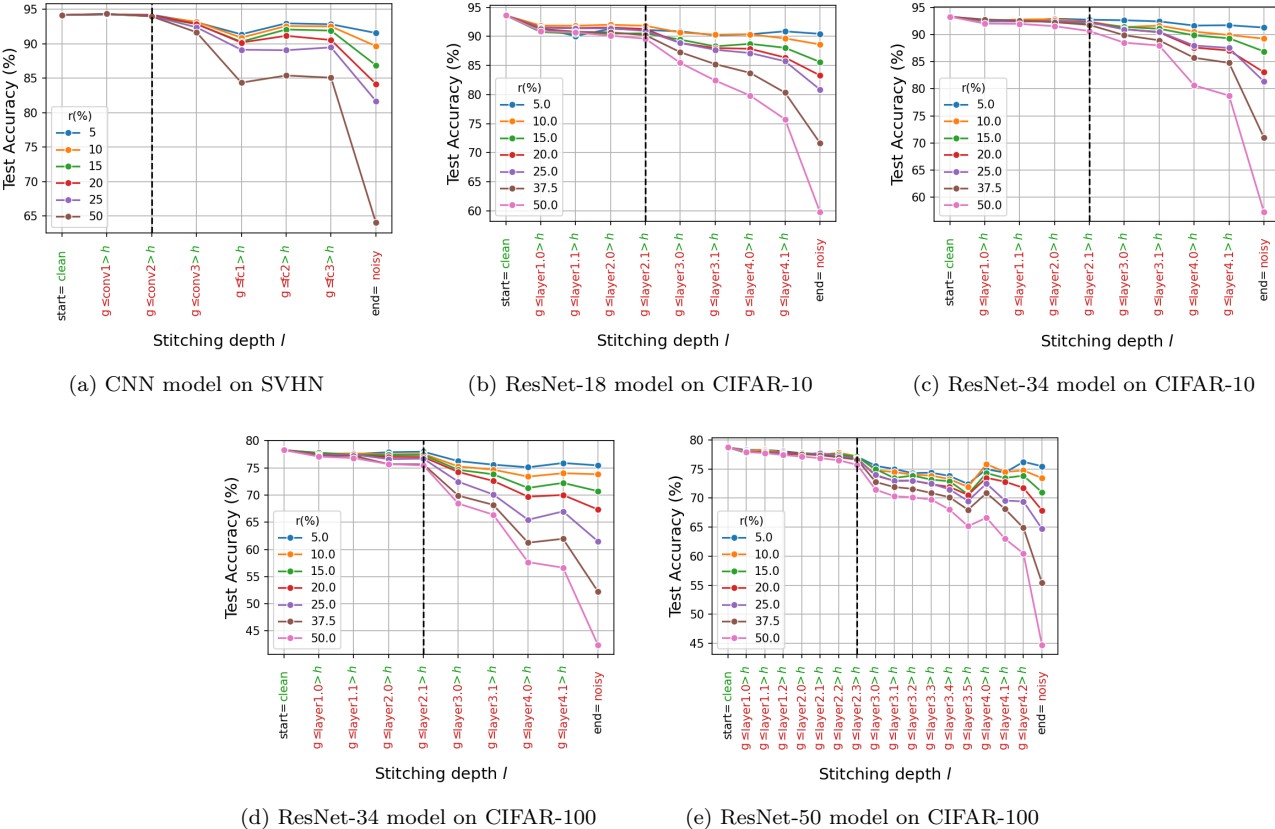

Figure 5: Noisy-to-Clean Stitching $h_0^{>l}(S(g_r^{\leq l}))$: Accuracy of stitched models on unseen test dataset. Each xtick corresponds to a stitched model comprising noisy base up to the layer $l$, stitching layer and the clean head starting from layer $l+1$ to the last layer. The black vertical line shows the minimum depth of the noisy base $l_n^g$ beyond which the generalization drops steadily in the clean model.

### 5.2.2 Noisy-to-Clean Stitching $h_0^{>l}(S(g_r^{\leq l}))$

We now consider the reverse direction: injecting early noisy (memorization-prone) layers into the clean model. Figure 5 presents the test accuracy of noisy-to-clean stitched models (y-axis) as a function of stitch depth $l$ (x-axis). The key observations are summarized below.

G.4 **Stitching a few early noisy layers causes minimal degradation.** After stitching in a few early noisy layers, the test performance of the stitched model remains close to that of the clean model. This indicates that a limited number of noisy bottom layers does not immediately harm generalization, likely because the majority of the upper clean layers can compensate or override the noisy representations introduced at shallow depths.

G.5 **Stitching beyond the minimum noisy prefix $l_n^g$ degrades performance** As stitching depth increases, test performance initially changes little, but then exhibits a sharp drop once the stitched noisy prefix exceeds a critical depth $l_n^g$. This $l_n^g$, referred to as the *minimum noisy depth that affects generalization*, marks the point at which noisy early representations overwhelm the corrective influence of the clean upper layers. Beyond $l_n^g$, test accuracy steadily declines and approaches that of the fully noisy model. Moreover, across all architectures and noise levels $r$, $l_n^g$ consistently appears in the first half of the model. This suggests that only a small number of noisy bottom layers can be tolerated before generalization deteriorates. However, even when all layers except the final classifier are noisy, the stitched model generalizes noticeably better than the noisy model, with improvements as large as

15–20% at $r = 50\%$. This highlights the dominant role of even a single-layer clean head that can significantly boost generalization, even when the entire base is noisy.

## 5.3 Interpretations and Inferences

We summarize the key observations from stitching clean layers to noisy model and vice verse in Table 1.

Table 1: Summary of generalization and memorization behavior under clean-to-noisy and noisy-to-clean stitching.

| Stitching Direction | Generalization (Test Accuracy) | Memorization (Random-Label Accuracy) |
|---|---|---|
| **Clean → Noisy** $h_r^{>l}(S(g_0^{\leq l}))$ | - Early clean layers give large gains. 
 - Improves with a deeper clean base. 
 - Matches clean model after stitching depth $l_c^g$. | - Early clean layers drops accuracy. 
 - Forgets memorization after stitching depth $l_c^m$. 
 - Clean base suppresses noisy head. |
| **Noisy → Clean** $h_0^{>l}(S(g_r^{\leq l}))$ | - Shallow noisy base has little effect. 
 - Sharp decline beyond stitching depth $l_n^g$. 
 - Always better than the fully noisy model. | - Negligible for small noisy prefix. 
 - Emerges only after stitching depth $l_n^m$. 
 - Never reaches 100% due to clean head. |

### 5.3.1 Fixing noise in early layers improves generalization and reduces memorization in noisy models

Our experiments demonstrate that fixing early layers of the noisy model, starting from the very first layer/block (see observation G.1), improves its generalization. The test accuracy of the noisy model increases steadily as more clean layers are stitched in (obs. G.2) and saturates once the clean base reaches a critical depth $l_c^g$. A similar behavior is observed for memorization, where memorized accuracy decreases as clean early layers are introduced (obs. M.1,M.2) and reduces to near-random behavior once layers up to $l_c^m$ are fixed (obs. M.3). Collectively, these observations show that—even when noise exists only in the upper layers—the stitched model behaves similarly to the clean model on both unseen and memorized inputs once early noisy layers up to a certain depth are replaced.

We attribute this to the hierarchical structure of DNNs, where the deeper layers depend directly on the representations produced by earlier ones. Any abnormality or noise-induced distortion in the early layers propagates and amplifies with depth. Consequently, replacing noisy early layers with clean ones effectively removes the source of distortion, improving generalization while suppressing memorization.

### 5.3.2 Stitching in noisy early layers to the clean model results in memorization with reduced generalization

Replacing the first few clean layers with noisy ones up to a depth $l_n^g$ does not degrade generalization (obs. G.5, M.5), maintaining test accuracy close to the clean model and keeping memorization near random. This is mainly due to the presence of a larger number of clean upper layers that can compensate for the noisy base. However, once the depth of the noisy base exceeds $l_n^g$, memorization emerges and test accuracy decreases, approaching the performance of the fully noisy model. This shows that although the model can tolerate a small noisy base, excessive corruption in early layers imposes a substantial burden on remaining clean layers, causing generalization to drop.

However, stitched models with the noisy base and clean head never achieves perfect memorization (100% accuracy on randomly labeled training inputs), even when all layers except the last layer are noisy. Instead, depending on architecture and dataset complexity, memorized accuracy ranges from as low as 20% to at most 90%. This highlights the critical role of the head layers in enabling complete memorization. This observation is consistent with existing memorization mitigation techniques, many of which modify or regularize the final layers. At the same time, the fact that noisy-to-clean stitched models still exhibit some memorization shows

that every layer participates in memorization. While early layers contribute, perfect memorization appears to require noise across all layers—including the head.

### 5.3.3 Implicit bias of architecture and task complexity on critical stitching depths

We study the impact of architecture and task complexity on layer-wise memorization and minimum stitching depth needed to restore generalization or suppress memorization. We experiment with two different architectures on the same dataset - ResNet-18 and ResNet-34 models trained on the same noisy variants of CIFAR-10, and study the behavior of stitched models. In this case, task complexity is fixed, and the model capacity varies. We observe that although the positions of critical depths ($l_c^g$, $l_c^m$, $l_n$, and $l_n^m$) are broadly similar, the rate at which accuracy changes with stitching depth differs. For instance, while both architectures exhibit the same minimum noisy depth $l_n^m$ for memorization to emerge, the increase in memorized accuracy varies (see Figure 3).

Conversely, when studying the same architecture (ResNet-34) across datasets of different complexity (CIFAR-10 vs. CIFAR-100), we find that the change in performance with stitching depth follows a similar trajectory, but the location of critical stitching depths shifts. For instance, in ResNet-34, a noisy base up to *layer4.0* marks the emergence of non-trivial memorization on CIFAR-10, while it is pushed deeper to the last residual block *layer4.2* for CIFAR-100 (consistent with observations in ResNet-50 architecture for CIFAR-100). These patterns demonstrate that both architectural capacity and task difficulty jointly shape the positioning of critical depths and the rate of change of memorization and generalization performance.

### 5.3.4 Functional similarity between layers of the noisy and clean model

As the stitching depth in clean-to-noisy stitching increases, the test performance of the stitched model improves relative to the noisy model. This shows that clean layers exhibit better generalization. Conversely, in noisy-to-clean stitching, the test performance of the stitched model drops relative to the clean model after stitching beyond $l_n^g$, and the degradation becomes more prominent as stitching depth increases. This suggests that memorization-induced degradation compounds in depth. In essence, the collective functional similarity between layers of the clean and noisy models (as measured by the performance gap of the stitched model relative to the head model) decreases with depth. This further supports the idea that the generalization capability of the early layers in the noisy model is weaker than that of the clean model, highlighting how the negative impact of noise propagates through the network. Furthermore, these findings align with observations made by in-place fixing/reducing memorization in individual layers. All these experiments clearly mark the noise influence in early layers and how fixing it can reduce memorization and improve test accuracy.

## 6 Discussion and Future Work

Using model stitching, we have analyzed the learning dynamics of each layer of DNN trained with random-label noise. Our empirical analysis both corroborates earlier findings in the literature as well as provides new insights. In particular, our experiments show that early layers are infected by training noise, and this is cascaded further in the DNN model owing to its feedforward nature. Furthermore, across datasets and models, we consistently observe that fixing early noisy layers can improve the model's performance on unseen data and reduce the accuracy on memorized inputs, even when noisy deeper layers remain untouched. To further verify and corroborate our findings, we perform another experiment (see Appendix A.1) where we fix noise in each layer of the model independently and then evaluate the performance of the model on test and memorized inputs. We also experiment with the insertion of clean middle layers/blocks in the noisy model via two stitching layers to study the sole influence of middle layers on the model's functional behavior in Appendix A.4. We once again observe that training and fixing noise in early layers do result in substantial improvement in generalization and suppresses memorization, corroborating that noise memorization starts early in the model. Furthermore, we show the dominant role of deeper layers for perfect memorization, and the influence of architecture bias and dataset complexity on the extent of memorization across layers. In the future, we plan to extend our work to transformer-based VLMs to uncover memorization dynamics between transformer layers and leverage our findings to develop novel methods to mitigate memorization.

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

## A   Appendix

### A.1   Retraining each layer of the model

So far, we assumed access to a noise-free clean model and compared the functional similarity of its layers to those of the noisy model. However, in a practical setting, it is more feasible to correct layers/blocks of the model by retraining than to replace them with those of the noise-free clean model. So, we raise a key question: "What would be the impact of in-place fixing/reducing noise in each layer/block of the noisy model on its performance, and whether observed findings would be similar to those of stitching noise-free layers" However, answering this question needs a method to fix layer-wise noise in the model, which itself is challenging and beyond the scope of our work. As a result, we adopt the popular approach of fine-tuning each layer on a dataset $\mathcal{D}_{ft}$ and evaluate the test and memorized accuracy of the end model.

We use the validation dataset $\mathcal{D}_{val}$ as the fine-tuning dataset $\mathcal{D}_{ft}$ for each layer, resulting in weight updates to learn more generalizable features. This way, we can assume reduced noise in the fine-tuned layer when trained on $\mathcal{D}_{val}$. So, after fine-tuning each layer independently (while freezing others), the model's performance is evaluated on the unseen test dataset $\mathcal{D}_{test}$ and memorized inputs $\mathcal{D}_r^n$. We experiment on the ResNet-18 model trained on multiple noisy variants of CIFAR-10 with noise ratio $r$ and report our findings in Figure 6.

The observations can be summarized as

- Retraining the very first block on $\mathcal{D}_v$ improves test accuracy and starts predicting correct labels to memorized inputs, and the increase in accuracy is proportional to the noise ratio $r$. This corroborates that noise impacts early layers, and its influence is directly proportional to the underlying noise ratio $r$.

- As the depth of the retrained layer increases, more memorized inputs are correctly predicted. However, we do not observe more than 80% accuracy on memorized inputs, showing limitations of fine-tuning in mitigating underlying noise and the limited corrective power of each layer.

Next, we repeat the above experiment using memorized inputs but with their true labels as the fine-tuning dataset, i.e., $\mathcal{D}_{ft} = \mathcal{D}_r^n$. The fine-tuning will directly interfere with the learned noise, providing the right update direction for weights to forget random labels and correctly predict ground truth labels on the memorized inputs, resulting in reduced noise. We again train ResNet-18 on varied noisy variants of the CIFAR-10 dataset and report their training and test accuracy in Figure 6. The experiment reveals that:

- The fine-tuning of the first block can map at least 70% of previously randomly labeled inputs to correct labels, despite no change in the rest of the model. Moreover, the accuracy with respect to true labels increases with the depth of the fine-tuned layer.

- Fine-tuning *layer3.0* or any layer after it achieves 100% accuracy on memorized inputs. Notably, *layer3.0* is also the minimum clean suffix needed to mitigate memorization completely (see Figure 2).

- Fine-tuning with correct labels of memorized input shows degradation in test accuracy for all noise ratios $r \leq 8000$. However, for large noise, the test accuracy improves drastically after fine-tuning the first layer and increases with an increase in depth of the fine-tuned layer, except for the last residual block.

We hypothesize that when a layer is fine-tuned with $\hat{\mathcal{D}}_n^r$, the gradient updates weights to forget memorized inputs and learn their true class label, messing up with existing generalized features. Moreover, during training on $\mathcal{D}_r^n$, the quality of learned generalized features is better if the noise ratio is small. That's why, for small noise ratios, fine-tuning early layers can learn generalized features that can negatively impact previously learned generalized features, while for large noise ratios, the quality of generalized features was already poor, and fine-tuning can partially improve them. This explains the drop in test accuracy for small $r$ and continuous improvement for large $r$.

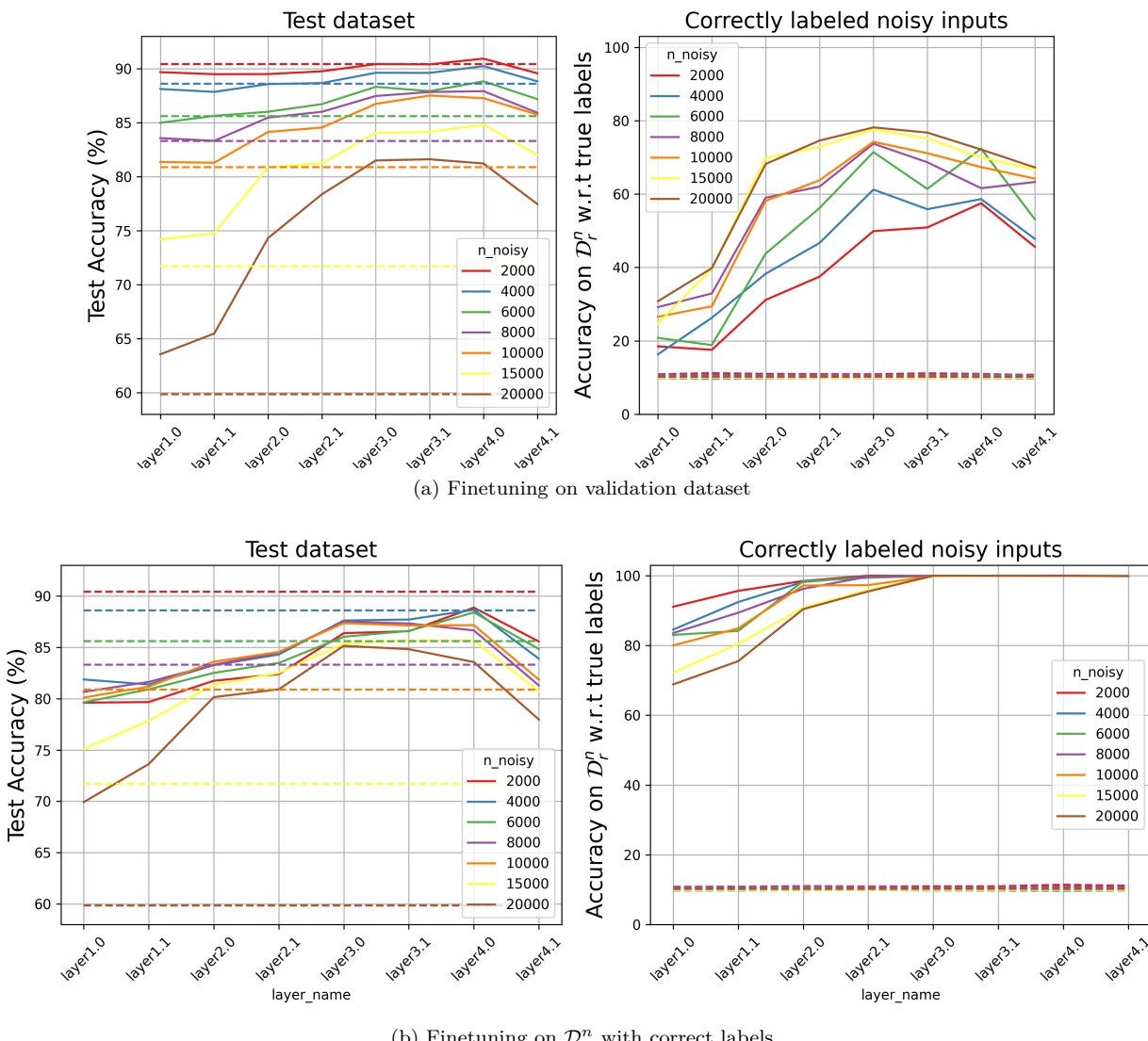

(a) Finetuning on validation dataset

(b) Finetuning on $\mathcal{D}_r^n$ with correct labels

Figure 6: **ResNet-18 on CIFAR-10**: The first row shows the performance of the model whose layers are independently trained on the validation dataset, and the second row corresponds to model layers being finetuned on the correctly labeled memorized inputs. In each subplot, all the residual blocks to be finetuned independently are listed on the x-axis while the model's accuracy (with exactly one finetuned block) is plotted on the y-axis. The first column shows the performance of the model with finetuned layers on the unseen test dataset, and the second column corresponds to the model's performance on $D_r^n$ with respect to their true class labels.

## A.2 Layer-wise study of memorization in very deep networks

We apply stitching to ResNet-101 to study the layer-wise memorization and generalization in very deep networks. ResNet-101 comprises of 33 residual blocks organized into four stages - `{layer1, layer2, layer3, layer4}` with `layer3` consisting of 23 residual blocks. To probe layer-wise effects at different depths, we perform stitching at representative locations: the first residual block of each stage (`layer1.0`, `layer2.0`, `layer3.0`, `layer4.0`) and the final residual block (`layer4.2`). The results of stitching are shown in Figure 7. Despite `{layer1, layer2}` forming a small fraction of the total network depth, stitching a clean prefix up

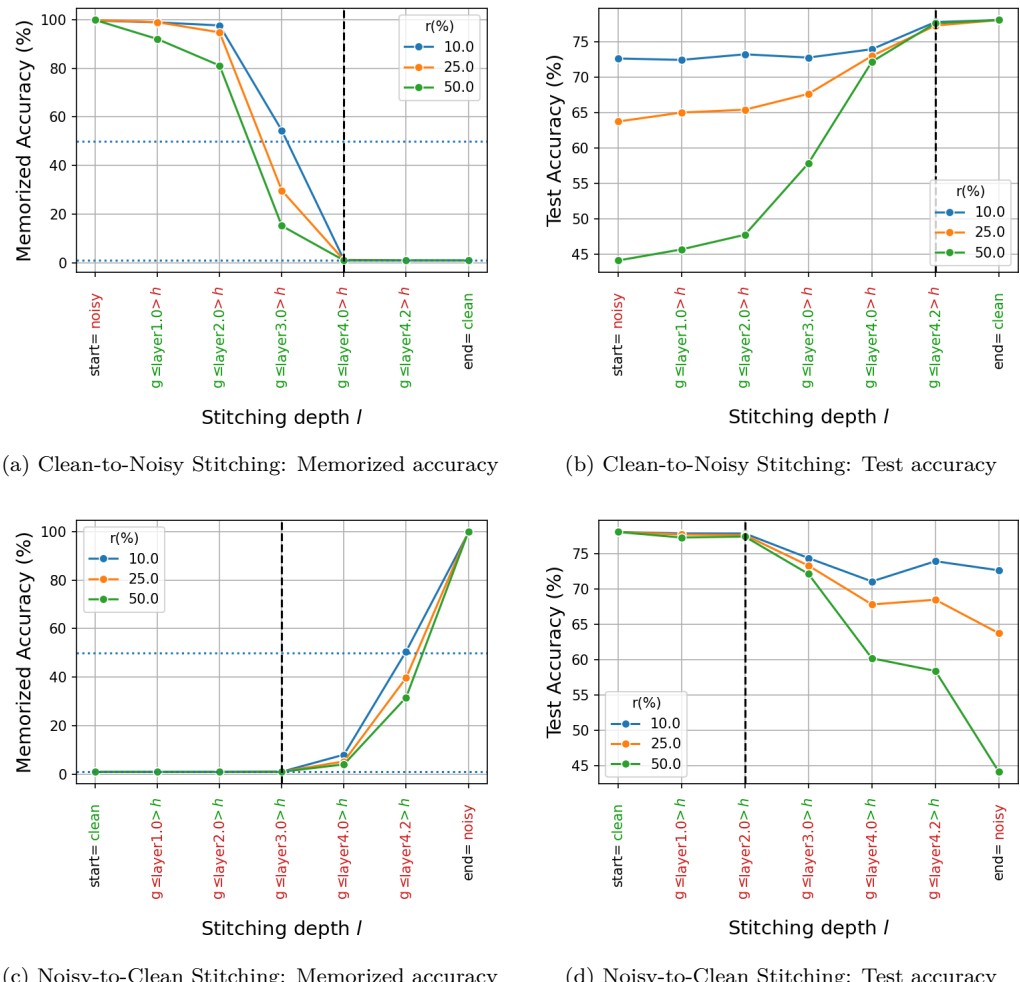

(a) Clean-to-Noisy Stitching: Memorized accuracy

(b) Clean-to-Noisy Stitching: Test accuracy

(c) Noisy-to-Clean Stitching: Memorized accuracy

(d) Noisy-to-Clean Stitching: Test accuracy

Figure 7: ResNet-101 trained on CIFAR-100.

to `layer3.0` eliminates more than 50% memorization for large noise ratios. In fact, the rate of reduction in memorized accuracy slows down as we stitch beyond `layer3.0`. Conversely, the emergence of memorization in noisy-to-clean stitched models is delayed until we stitch the noisy prefix up to the last few residual blocks. In fact, the coarse stitching behavior, both clean-to-noisy and noisy-to-clean, of ResNet-101 seems to resemble that of ResNet-34 and ResNet-50 trained on the CIFAR-100 dataset. The observed functional behavior of stitched models (evaluated as memorized and test accuracy) is consistent with all our findings.

### A.3 Layer-wise study of memorization in Transformer model

In this paper, our analysis is intentionally scoped to convolutional and residual architectures, where the notion of locality and hierarchical feature reuse allows a controlled investigation of memorization mechanisms. To investigate whether the observations also generalize to other architectures, we conduct preliminary experiments on Vision Transformers (ViT). Specifically, we train a ViT-Small architecture on CIFAR-100 with and without random noise. We then stitch clean model ($r = 0$) to noisy model ($r \in \{25\%, 50\%\}$) after each transformer block, considering both clean-to-noisy and noisy-to-clean directions. The stitching layer is trained on the validation dataset, and evaluation of stitched models on the unseen test dataset and memorized randomly labeled inputs is plotted in Figure 8. The key observations are:

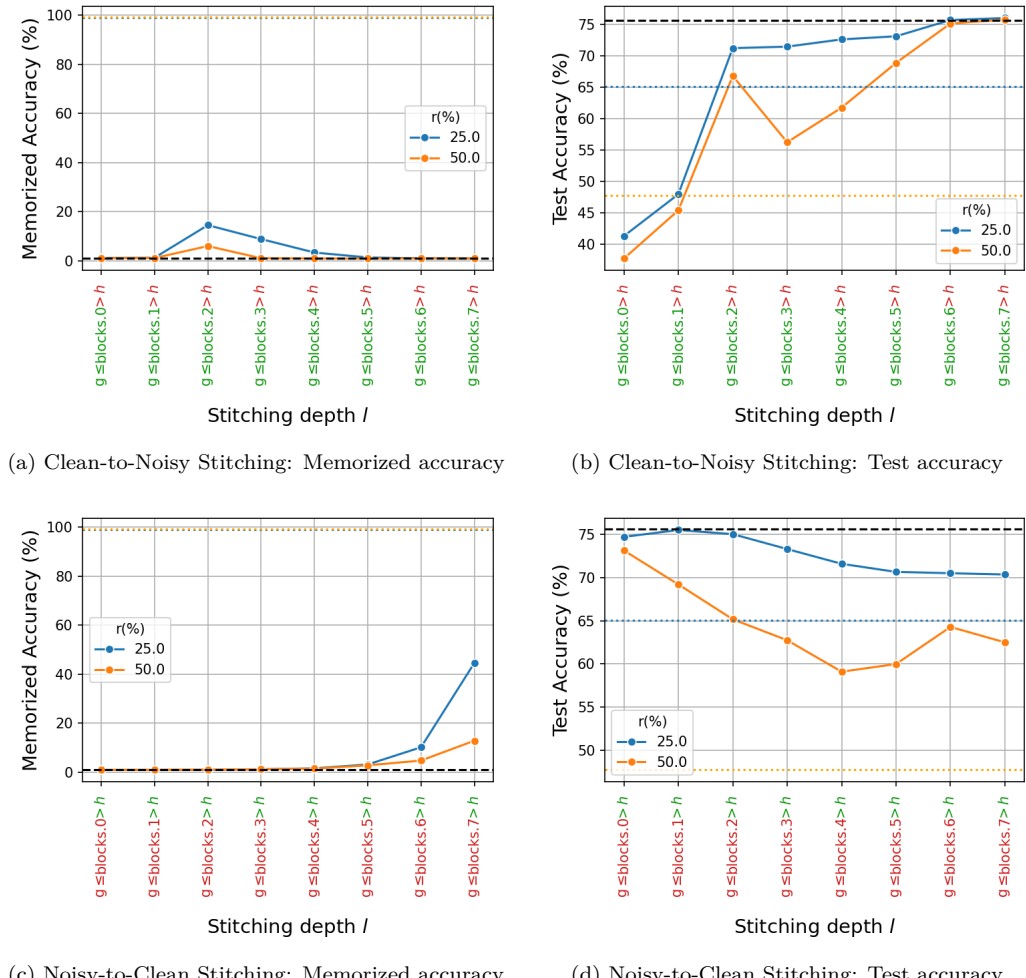

(a) Clean-to-Noisy Stitching: Memorized accuracy

(b) Clean-to-Noisy Stitching: Test accuracy

(c) Noisy-to-Clean Stitching: Memorized accuracy

(d) Noisy-to-Clean Stitching: Test accuracy

Figure 8: Vision Transformer (ViT-s) trained on CIFAR-100. The dotted lines in the first column show the memorized accuracy of noisy models. The dotted lines in the second column show the test accuracy of noisy models. The black dashed line shows the memorized and test accuracy of the clean model.

- Clean-to-Noisy Stitching: Stitching a clean base to the noisy model reduces memorization at all stitching depths except `blocks.3`, where a sudden, small spike in memorized accuracy is observed. This behavior is broadly consistent with our earlier findings M.1. Furthermore, stitching a sufficient number of initial clean transformer blocks to the noisy model improves its test performance. In this case, stitching layers of clean model at least up to `blocks.2` improves the test accuracy of the noisy model for both noise ratios ($r = 25\%$ and $r = 50\%$). Also, once the clean model is stitched upto `blocks.6`, the test performance of the stitched model matches that of the clean model irrespective of noisy head layers and noise percentage $r$. This is consistent with observed findings G.1 and G.2.

- Noisy-to-Clean Stitching: As early noisy layers are stitched into the latter clean layers, the test accuracy starts decreasing. For a smaller noise ratio ($r = 25\%$), the drop in test performance is gradual, while a sharp decrease is observed for a larger noise ratio ($r = 50\%$). However, irrespective of the stitching depth (i.e., the number of noisy blocks stitched into the clean model), the test accuracy of the stitched model is always significantly better than that of its noisy counterpart. This aligns with the well-known fact of fixing the last few layers of the noisy model to improve its test performance. Interestingly, the memorization emerges only when a significant number of initial noisy blocks are

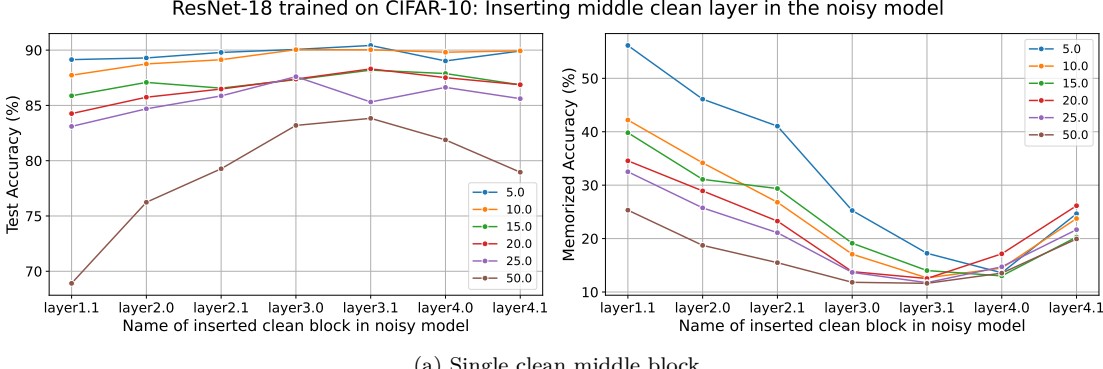

(a) Single clean middle block

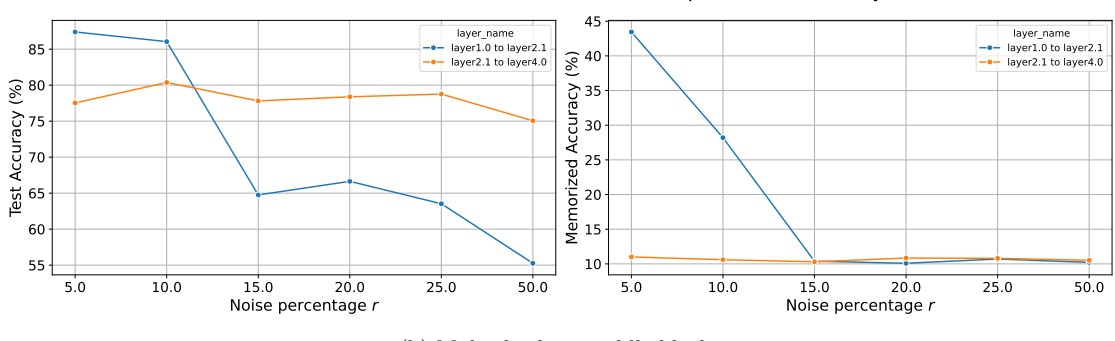

(b) Multiple clean middle blocks

Figure 9: Inserting clean middle blocks in the noisy model. First row corresponds to insertion of single clean middle blocks, and the second row corresponds to insertion of multiple clean middle blocks in the noisy model.

stitched into the clean model, and perfect memorization is never witnessed even when all the layers except the last output layer are noisy, consistently aligning with our main findings M.4 and M.5.

Despite these qualitative similarities, ViT stitching also exhibits deviations from observed patterns in CNN and ResNets. For instance, the sudden increase in memorized and test accuracy when the clean model is stitched to the noisy model at `blocks.2` (subplots (a) and (b) in Figure 8). Similarly, the test accuracy of the noisy-to-stitched model for a large noise ratio $r = 50\%$ drops sharply with stitching in early noisy blocks and then fluctuates as more noisy blocks are stitched in. These irregularities warrant further investigation to determine whether they stem from architectural differences or from limitations of the stitching mechanism itself, as, unlike convolutional models, stitching in ViTs applies a token-wise linear transformation. Overall, our preliminary experiments suggest qualitatively similar trends with minor deviations, but a full analysis would require additional methodological development and is beyond the scope of the present work.

## A.4 Insertion of middle clean layers in the noisy model

Model stitching stitches a clean prefix to the noisy model, fixing noise in all the layers of the prefix. As a result, understanding the noise impact of each layer independently or a subset of layers other than the prefix is unaddressed. So, to delve further into noise memorization by each layer, we replace the middle blocks of the noisy model with their clean counterparts. This is achieved by inserting two stitching layers, i.e., first performing noisy-to-clean stitching at layer $l_1$ followed by stitching of the stitched model to the noisy model at layer $(l_2)$, resulting in swapping out of noisy layers $(\{l_1 + 1, \ldots, l_2\})$ by their clean counterparts. We carry out experiments on ResNet-18 trained on CIFAR-10 for all noise ratios $r$ by a) inserting one middle clean block in the noisy model and b) inserting a consecutive subset of clean middle blocks in the noisy model. ResNet-18 consists of eight residual blocks, with seven blocks that can be considered "middle" blocks, leading to many

| $r$ (%) | layer1.0 | layer1.1 | layer2.0 | layer2.1 | layer3.0 | layer3.1 | layer4.0 | layer4.1 |
|---|---|---|---|---|---|---|---|---|
| 5.0 | 48 | 62 | 98 | 113 | 206 | 237 | 432 | 497 |
| 10.0 | 49 | 62 | 110 | 113 | 206 | 244 | 431 | 494 |
| 15.0 | 55 | 60 | 98 | 113 | 184 | 233 | 394 | 486 |
| 20.0 | 51 | 62 | 106 | 113 | 201 | 240 | 355 | 473 |
| 25.0 | 50 | 62 | 108 | 113 | 188 | 224 | 434 | 484 |
| 37.5 | 55 | 62 | 98 | 113 | 182 | 244 | 369 | 476 |
| 50.0 | 49 | 60 | 95 | 113 | 165 | 229 | 259 | 468 |
| Maximum rank | 64 | 64 | 128 | 128 | 256 | 256 | 512 | 512 |

Table 2: Noisy-to-clean stitching: Rank of stitching layers

| $r$ (%) | layer1.0 | layer1.1 | layer2.0 | layer2.1 | layer3.0 | layer3.1 | layer4.0 | layer4.1 |
|---|---|---|---|---|---|---|---|---|
| 5.0 | 59 | 61 | 95 | 114 | 188 | 236 | 427 | 496 |
| 10.0 | 59 | 61 | 95 | 114 | 188 | 244 | 427 | 498 |
| 15.0 | 59 | 59 | 95 | 114 | 188 | 233 | 403 | 496 |
| 20.0 | 59 | 63 | 95 | 114 | 188 | 239 | 363 | 488 |
| 25.0 | 59 | 63 | 95 | 114 | 188 | 223 | 427 | 497 |
| 37.5 | 59 | 62 | 95 | 114 | 188 | 244 | 378 | 493 |
| 50.0 | 59 | 59 | 95 | 114 | 188 | 228 | 270 | 485 |
| Maximum rank | 64 | 64 | 128 | 128 | 256 | 256 | 512 | 512 |

Table 3: Clean-to-Noisy stitching: Rank of stitching layers

possible subsets. To keep the study tractable, we focus on insertion of two consecutive subsets of clean middle blocks - {layer1.1, layer2.0, layer2.1} (implemented by inserting stitching layers at `layer1.0` and `layer2.1`) and {layer3.0, layer3.1, layer4.1, layer 4.2} (implemented by inserting stitching layers at `layer2.1` and `layer4.2`). The functional behavior of the noisy model after insertion is shown in Figure 9. We observe that inserting clean middle blocks leads to a more pronounced reduction in memorized accuracy compared to replacing only early layers, consistently across all noise ratios $r$. Moreover, inserting larger consecutive subsets of middle blocks (three or four blocks) further mitigates memorization. However, a key distinction between stitching clean middle blocks vs clean-prefix stitching lies in generalization. Contrary to clean-prefix stitching, none of the explored insertions has resulted in the test accuracy similar to that of the clean model for any noise ratio. While this suggests that fixing middle layers alone may be insufficient to fully recover generalization, we refrain from drawing strong conclusions, as a comprehensive evaluation would require exploring a large number of possible block combinations. We leave such an exhaustive combinatorial analysis to future work.

## A.5 Analysing stitching layers

In our experimental study, the stitching layer connects the memorization-heavy component of the noisy model to the downstream layers of the noise-free clean model and vice versa. In such a scenario, the task-loss-based optimization can encourage discarding of noisy features that might be detrimental to the downstream task performance. As a result, even without any explicit regularizer enforcing low-rank, the task-based optimization can result in a non-invertible linear transformation if doing so maximizes the functional alignment between representations (marked by converging loss during training). So, to experimentally corroborate our hypothesis, we have computed the rank of the stitching layer (considering various depths across the network where the stitching layer could be inserted). Consistent with our expectations, the stitching layer indeed turns out to be a rank-deficient matrix across multiple stitching depths and noise levels. The Tables 2 and 3 report the rank of all stitching layers of the ResNet-18 model trained on CIFAR-10. The rows (except the last) correspond to noise level $r$ in the training dataset, and the columns correspond to the stitching depth – layers at which stitching is performed. The last row shows the maximum possible rank (size) of the matrix determined by

the layer dimensionality. This result demonstrates that rank-deficiency emerges naturally in stitching layers during training without any explicit regularization.

### A.6 Training plots of stitching layers

The stitching layer $S$ is trained by minimizing the cross-entropy loss of the stitched model on the held-out validation dataset. The training proceeds until loss converges, at which point the representations of the two models are maximally aligned in order to achieve the best-possible performance of the stitched model on its training (i.e., validation) dataset. The training convergence of stitching layers, both for clean-to-noisy and noisy-to-clean stitching, is shown in Figure 10, 11, 12, 13, 14 and 15.

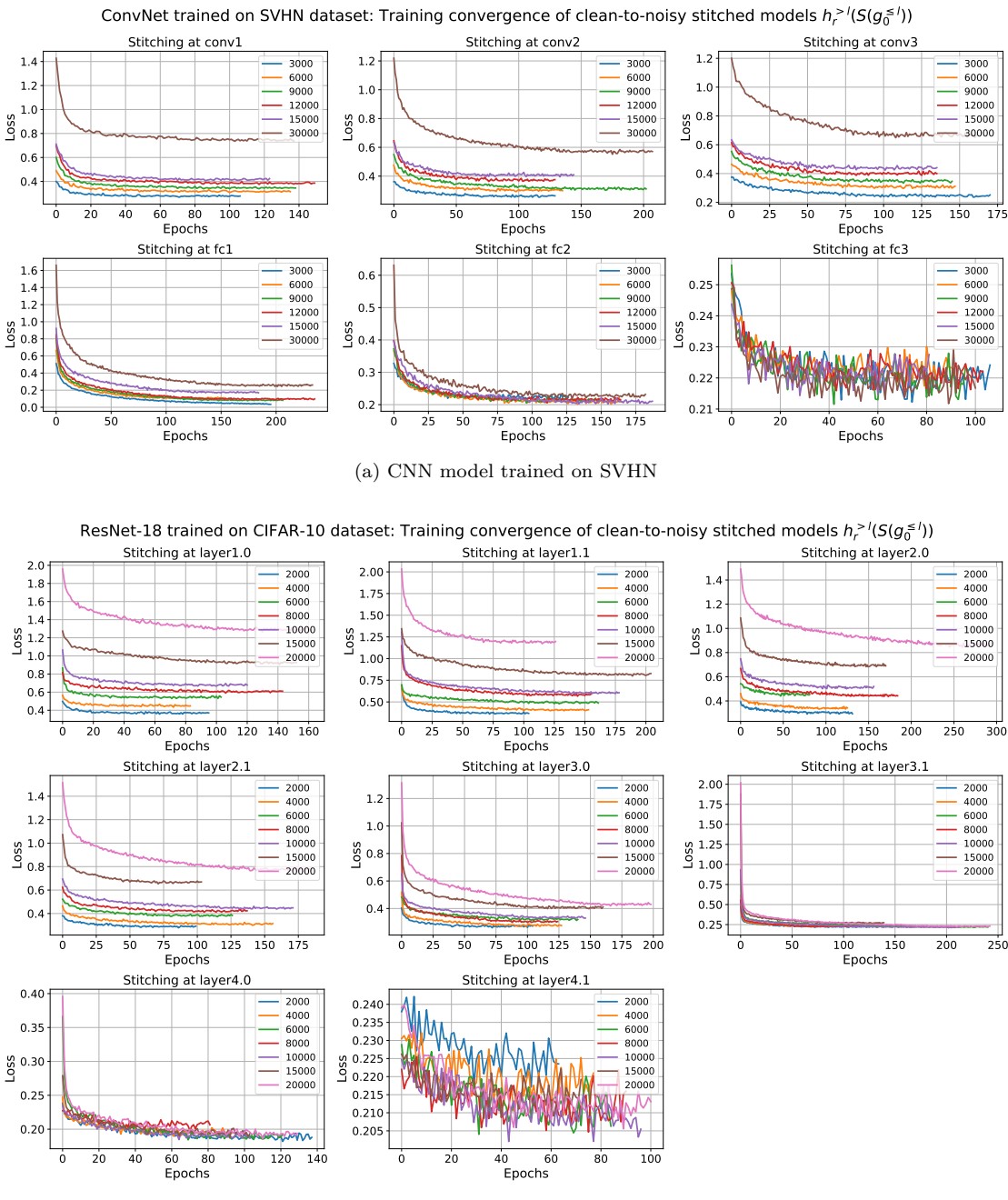

Figure 10: Training convergence of stitching layers in clean-to-noisy stitched models

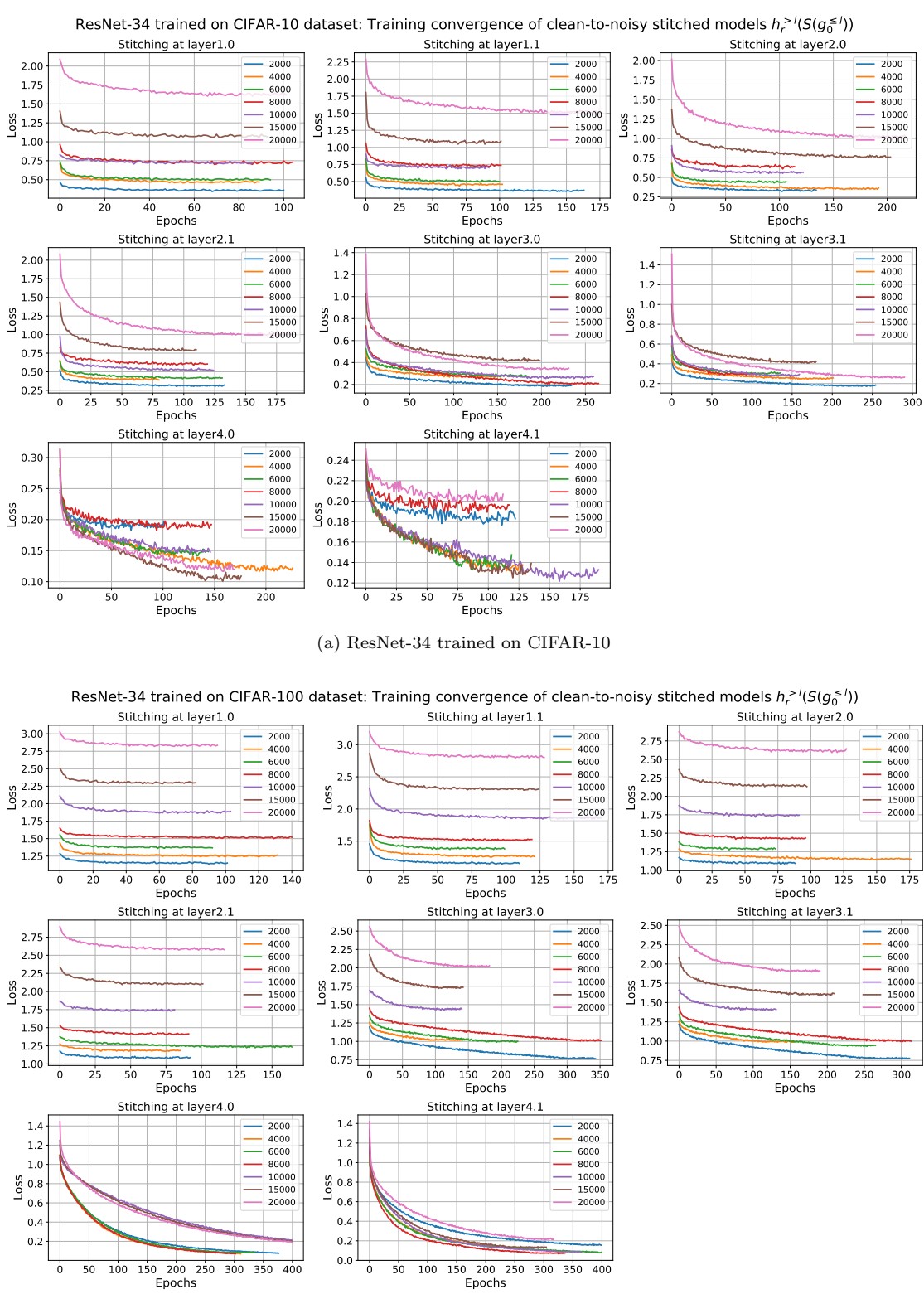

Figure 11: Training convergence of stitching layers in clean-to-noisy stitched models

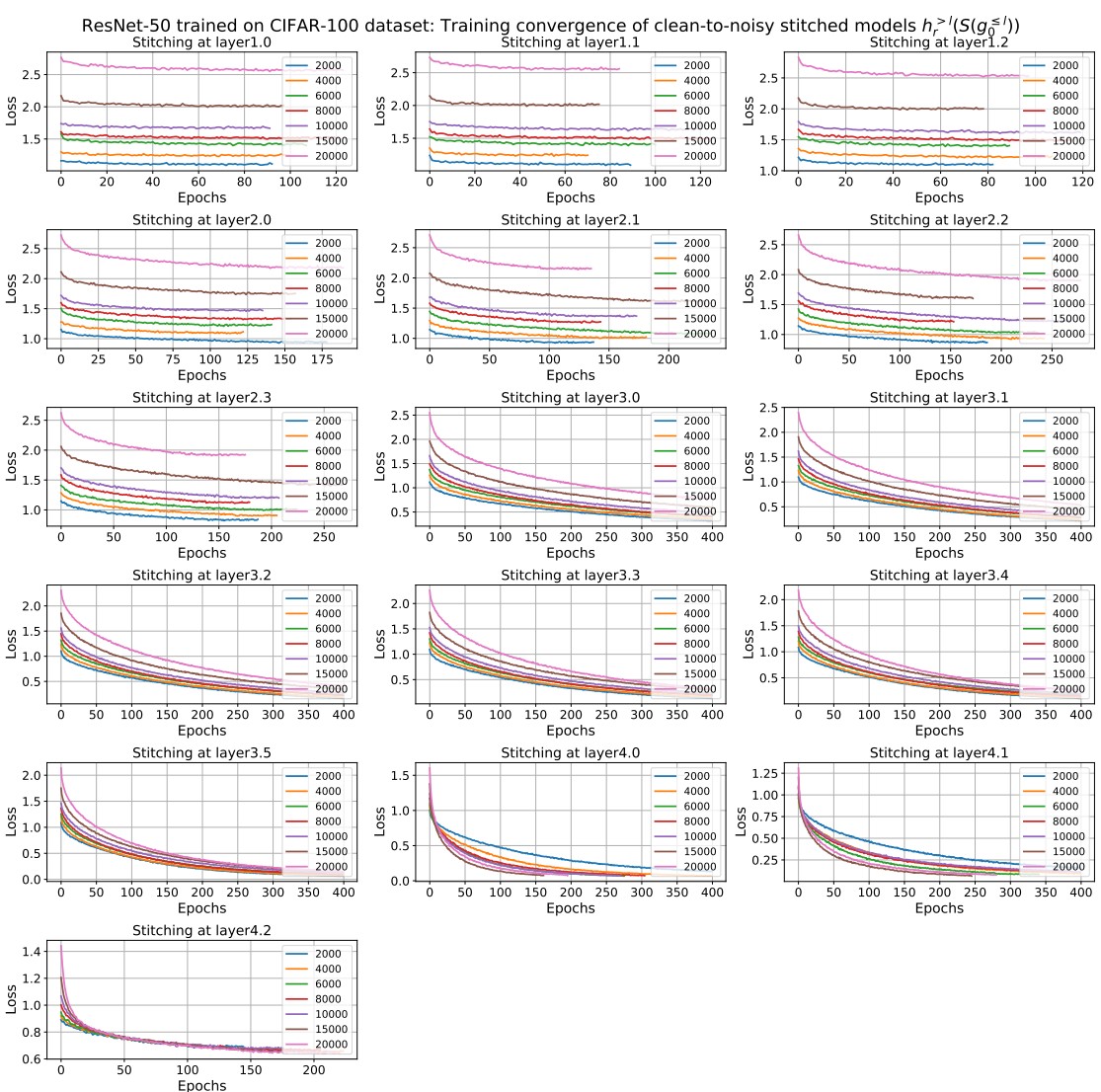

Figure 12: ResNet-50 trained on CIFAR-100: Training convergence of stitching layers in clean-to-noisy stitched models

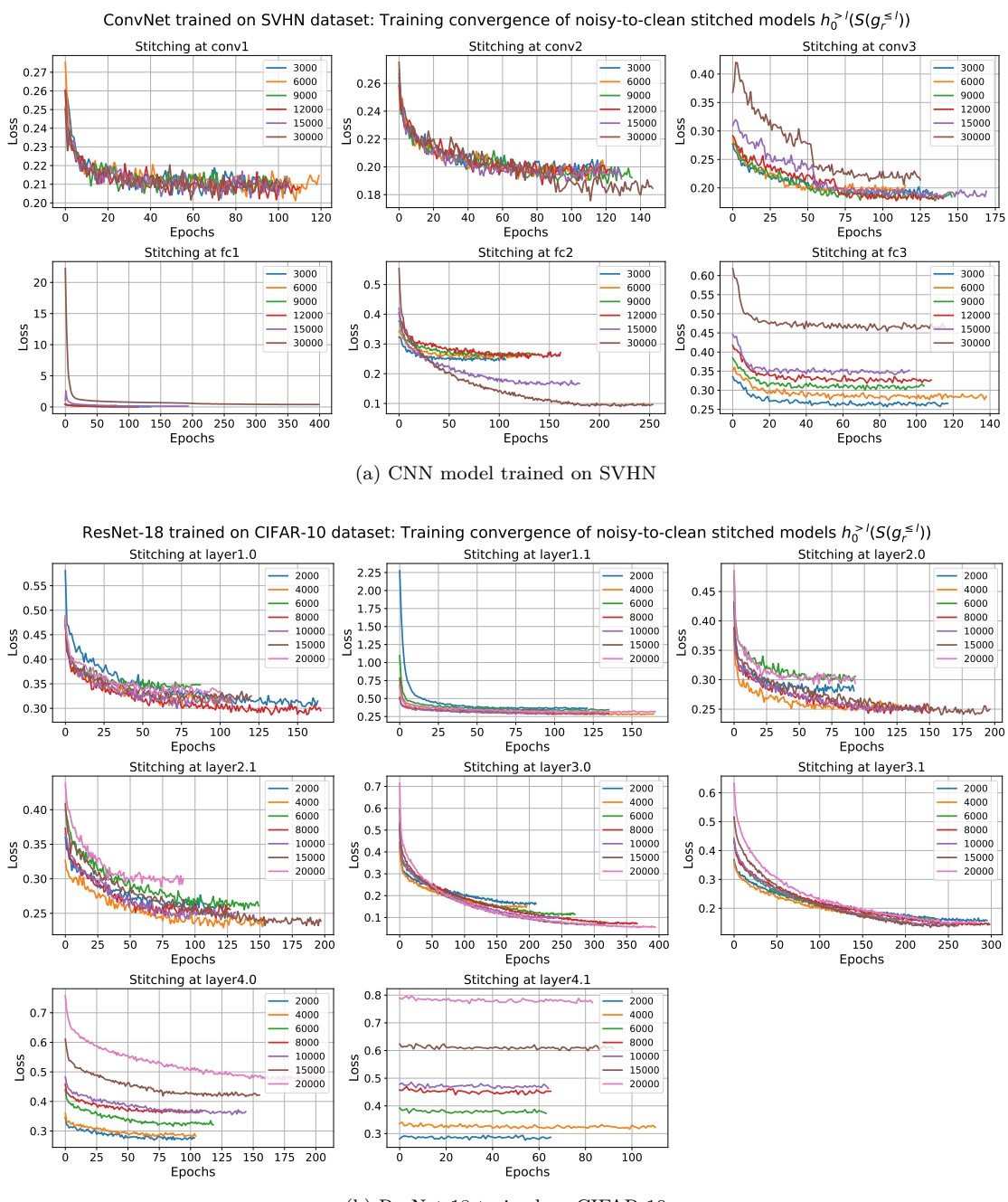

Figure 13: Training convergence of stitching layers in noisy-to-clean stitched models

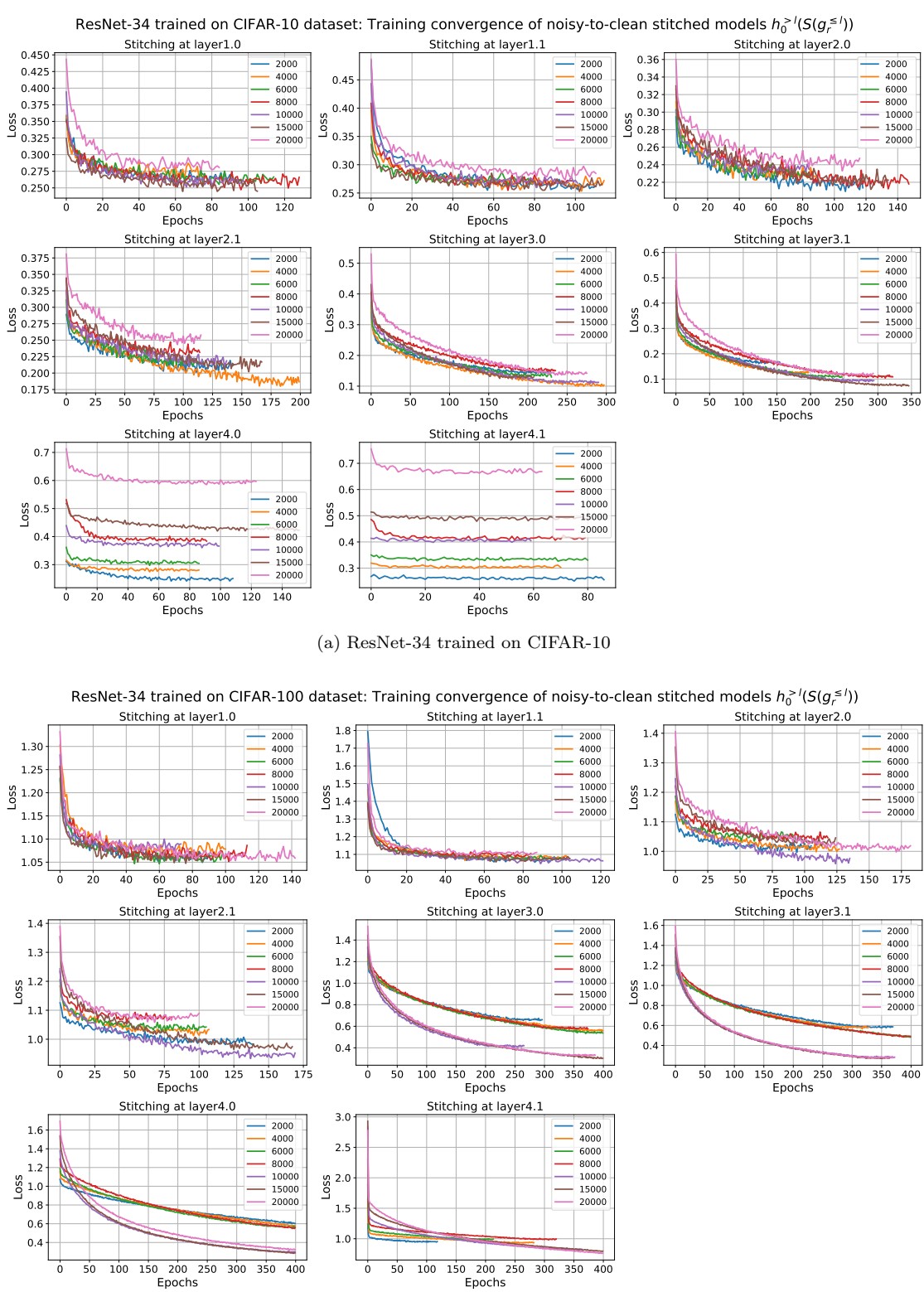

Figure 14: Training convergence of stitching layers in noisy-to-clean stitched models

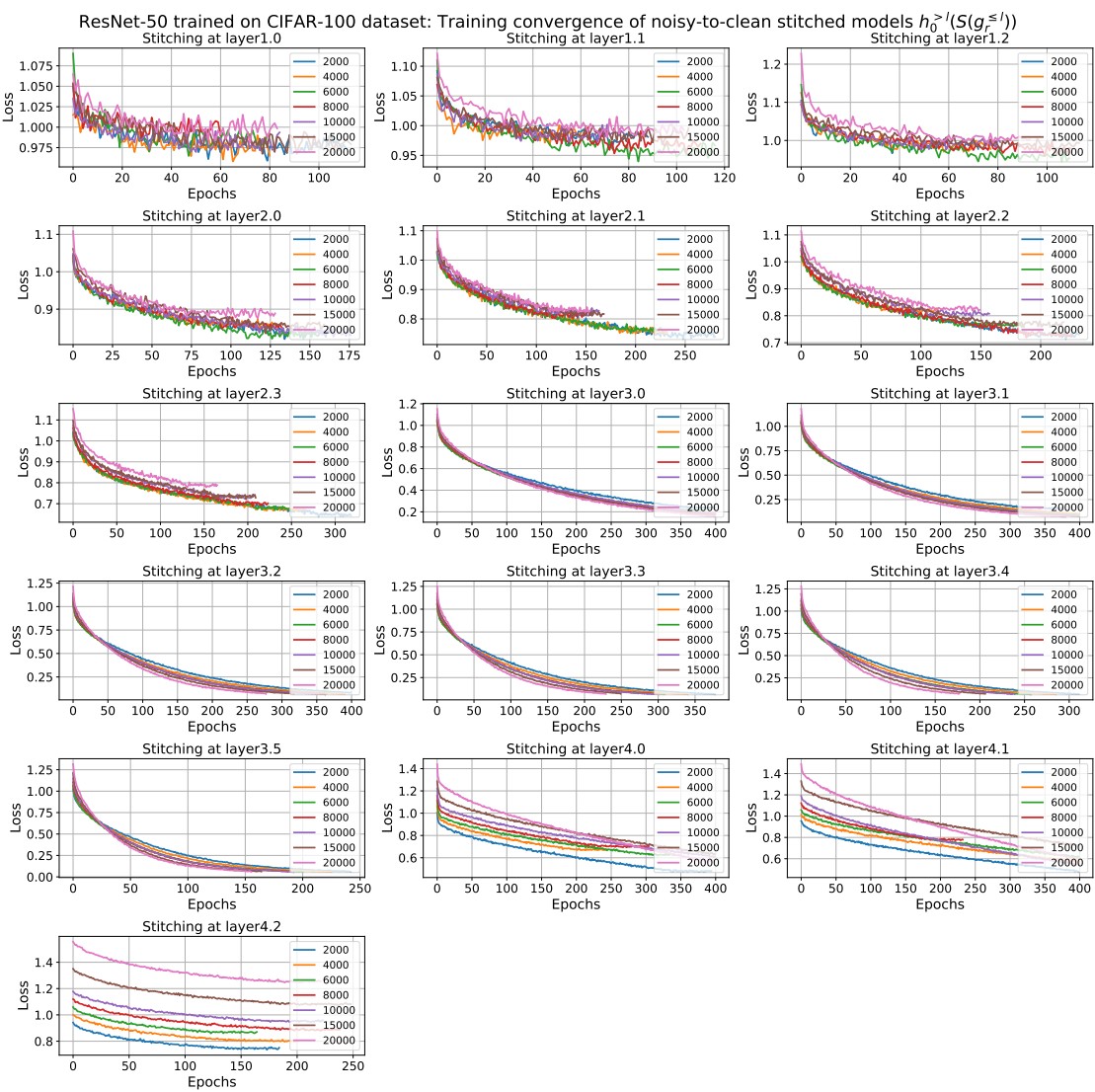

Figure 15: ResNet-50 trained on CIFAR-100: Training convergence of stitching layers in noisy-to-clean stitched models

