# OpenReview forum: "Probing Layer-wise Memorization and Generalization in Deep Neural Networks via Model Stitching"
_TMLR — Accepted by TMLR_

### Review · Reviewer_FKUA · 2025-12-14

**Summary Of Contributions:**

The paper introduces a method to investigate and localize the layers responsible for memorization in DNNs (specifically CNN, ResNet-18, and ResNet-34) via model stitching. The methodology relies on two distinct models: a "clean" model trained on a standard dataset and a "noisy" model trained on a dataset with randomly assigned labels. The authors explore two stitching directions: substituting clean layers into the noisy model (Clean-to-Noisy) and introducing noisy layers into the clean model (Noisy-to-Clean) .

Through these experiments, the authors provide a set of well-validated findings:
- Early Layer Susceptibility: Contrary to the assumption that early layers learn robust features, the study finds they are infected by noise; however, replacing them with clean counterparts effectively mitigates memorization .
- Asymmetry in Stitching: The authors demonstrate that the two stitching directions are asymmetric—cleaning early layers in a noisy model significantly restores performance, whereas introducing early noise into a clean model does not immediately degrade generalization .
- Role of Model Head vs. Base: The results highlight that while early layers contribute to noise propagation, the deeper layers (head) play a dominant role and are necessary for achieving perfect memorization.

The paper validates these claims using standardized classification datasets (CIFAR-10, CIFAR-100, and SVHN). Overall, the proposed approach offers solid empirical evidence regarding layer-wise memorization behavior in classification tasks.

**Additional Comments:**

The paper offers valuable insights into memorization dynamics over the classification task using a well-structured methodology, particularly the use of two-way model stitching between noisy and clean models. However, the reliance on prefix-based stitching introduces a potential bias that may overemphasize the role of early layers compared to intermediate ones. While I lean towards acceptance due to strong evidence presented for the technical findings, addressing the requested changes regarding interval-based evaluation, inclusion of backbones different in architecture and number of parameters, and broader task validation would increase confidence in the generalizability of the results.

**Audience:**

Yes

**Audience Explanation:**

The paper provides insights into one of the fundamental yet unresolved problems in deep learning: memorization of training data. Although the presented experiments involve relatively small datasets and models compared to the scale of current state-of-the-art systems, the observations provided are valuable. In particular, the identification that generalization and memorization are not symmetric behavior, demonstrated through the two-way stitching of models (between the clean and noisy models), is an important finding. Furthermore, I believe that the study offers broad insights that will benefit researchers attempting to solve the memorization problem, even if it is on scaled up models which still requires fundamental findings irrelevant from the model scale.

**Broader Impact Concerns:**

The proposed method focuses on investigating the memorization behavior on existing architectures with standardized datasets. Overall, the authors neither introduce a new model nor benchmark that may create a sensitive issue in terms of broader impact. However, since the proposed method can facilitate research on memorized sensitive content, any may possibly lead to unlearning approaches with a more careful block selection with the stitching methodology, it can even facilitate further research on existing broader impact concerns in the field.

**Claims And Evidence:**

Yes

**Claims Explanation:**

For each of the major observation provided in the paper, authors follow with a quantitative analysis on the classification datasets CIFAR-10, CIFAR-100, SVHN. Their approach is pretty standardized. Furthermore, the experimental design includes validations across architectures (CNN and ResNet) with different noise ratios (5% to 50%) for a robust evaluation protocol. The evidence is particularly convincing due to the careful choice of training the stitching layer on a held-out validation set, which prevents the model from simply learning the noise patterns(see the final paragraph in page 4). Additionally, the authors clearly support their findings through experiments in Appendix A.1 (layer-wise retraining), where each layer is trained with a different noise ratio. Overall, the proposed method is well-validated and convincing.

**Requested Changes:**

While the submission provides important insights on the memorization with the classification task, there are certain changes that I believe to be beneficial for the manuscript. Please find the suggested changes below:
- **Evaluation of Intermediate Layer Intervals:** The current methodology relies exclusively on prefix-based stitching (e.g., replacing layers $0 \dots l$ with clean counterparts). This design inherently biases the analysis towards the importance of preceding layers, as they are always the first to be cleaned in every configuration. To verify the claim that early layers are uniquely susceptible to noise, I believe experiments on distinct layer intervals, specifically, replacing intermediate blocks would be beneficial and would strengthen the argument on the significance of the preceding layers (e.g., keeping the initial layers noisy, cleaning the middle blocks, and keeping the head noisy). This is necessary to determine if the observed benefits stem specifically from fixing the early layers or simply from fixing any significant portion of the network.
- **Validation on Larger Models:** While the experiments with ResNet-50 are appreciated, modern architectures often extend significantly deeper even for the classification task. From this point of view, I believe deeper architectures, including deeper backbones such as ResNet-101 or ResNet-152 and different architectures such as ResNeXT would be beneficial. It is important to verify if the "early layer" effect persists when the initial layers constitute a much smaller fraction of the total network depth, or if the noise becomes more distributed in significantly deeper networks.
- **Generalization to more Complex Tasks (e.g., Object Detection):** The current findings are limited to relatively simple classification tasks on SVHN, CIFAR-10, and CIFAR-100. To demonstrate the broader applicability of these memorization dynamics, evaluations on more complex tasks, such as Object Detection (e.g., with COCO). Memorization dynamics may differ significantly when the model must predict bounding boxes rather than just class labels, and this would clarify if the observed asymmetry is a fundamental property of DNNs or specific to classification.

---

### Review · Reviewer_AVA2 · 2025-12-15

**Summary Of Contributions:**

The paper studies the DNNs ability to generalize and memorize the training dataset using lens of functional representations similarity. Specifically, the authors apply layer-stitching method to better understand the (representational) differences and similarities of the models trained on the standard (noise-free) datasets and their noisy counterparts.

Key strengths:
- Important topic
- Representational perspective offers a universal angle to look at the models, thus experiments could be easily applied to different scenarios (architectures, datasets, modalities)
- Overall presentation and the scope of the work is well defined. The authors do not try to touch too many aspects at once) and clearly state the contributions of the work which are supported by the empirical evidence.

Weaknesses:
- Narrow experimental scope -- although the authors claim to use four different architectures and three different datasets, I would argue that using different sizes of ResNets shouldn't be considered as a different architecture. Regardless of the size (ResNet-18/34/50) these models share the core building blocks. This makes me wondering how broadly the findings would apply when considering size-homogenous models (like Transformers) or skipless-models like VGGs or MLPs.
- Although the overall presentation of the material is rather clear, I find it difficult to follow the specific structure of the experimental section. I would suggest the authors to consider different presentation of the findings. In particular, I’d encourage the authors to focus on two cases (Noise -> Clean (evaluated both on Memorization and Generalization tasks) and Clean -> Noise evaluated the same way. This would give a better perspective on how stitching particular subsets of layers affect model’s performance).
- The whole analysis relies only on the stitching approach. I think that at least partial control-experiments with some other types of functional similarity should be run to understand whether the findings are robust or just specific to stitching approach.
- Some (minor) claims made by the authors remain speculative and need further justification or explanation to be clear (I will list them in the later sections).

**Audience:**

Yes

**Audience Explanation:**

I believe so, the topic is still an important research direction with lots of questions unanswered and the work tries to offer novel perspective on the topic. At the same time, the limited experimental scope makes me doubt the the current form is ready for publishing the work.

**Broader Impact Concerns:**

I have no ethical concerns about this work.

**Claims And Evidence:**

Yes

**Claims Explanation:**

I voted yes, as the core claims are generally supported by the experiments (although the scope of the experiments is relatively narrow as I mentioned earlier).

However, the paper has multiple sentences which are not clear to me and would require further elaboration. I expect that some of these claims might not be obviously true for the rest of the readers of the paper so I’d encourage the authors to elaborate a little bit on these things.

In several places (p. 3, p. 4, p. 5) the authors mention that stitching works by “aligning the representations”. Specifically, in p.4 the authors write that stitching works by“ (...) maximizing the alignment between latent representations of two models”. What do the authors actually mean by “alignment” here? And could the authors actually show that training the stitching layers leads to maximization of that alignment (however measured it is?).
“We train the model on clean data without label noise so that none of its training inputs to be specifically memorized, and use it as a baseline for generalization.” While I understand the experimental design here and the need to have a reference of a clean dataset, I have to stress the fact that even the standard datasets with correct labels are not noise-free. It is a well-known fact that in most of the image classification datasets there is a clear hierarchy of learning easy to hard samples, where the hard ones are frequently ambiguous ones leading to a noise in the overall dataset.
Stitching vs. CKA -- the authors write that “Empirical evidence shows that even when two networks’ representations score low on metrics like CKA, they can still exhibit high functional similarity, demonstrating that representational and functional similarit y capture the complementary aspects of how representations relate.” The work cited by the authors indeed proposed an experiment (Section 6.3) where the representational similarity was decreased (around 0.7 CKA score) yet the accuracy of the stitched model remained unchanged. In this particular experiment, the authors regularized the stitching layer to decrease the CKA and keep the overall performance of the model unchanged -- such a scenario would likely never occur in real setting. Thus, in my opinion, this is not a sufficient evidence to argue that actually the functional similarity (measured with stitching) gives a different perspective than the one obtained with CKA or other types of representational similarity. Since the CKA/CCA metrics are invariant to reversible linear transformations, I would actually expect the stitching (through a linear layer) method and CKA/CCA scores to behave similarly. Such an analysis would definitely strengthen the work and help us better relate it to other findings studying the problem of memorization/generalization in a broader context.
“(...) the stitching layer cannot align the divergent representations (...)” -- again what do you mean by “divergent” representations and how do you make sure that this in fact cannot happen (i.e. these representations cannot be aligned)?

**Requested Changes:**

Here are some minor details I would encourage the authors to apply to generally improve the readability of the work. At the same time I'd like to repeat my suggestions about restructuring the results section to ease the overall readability.


Figure 1 is in poor quality -- upload this as a PDF instead of jpg.

I believe that Figure 2 is inconsistent in terms of the placement of black vertical dashed lines. Per Figure caption these lines “(...) shows the minimum depth of clean base layer required to mitigate memorization in noise models” however in case of ResNet-34 on CIFAR-100 and ResNet-50 on CIFAR-100 the dashed lines are placed at layers which still have almost perfect memorization. Am I missing something? Actually the analogous thing happen in Figure 4 again for ResNet-34 and ResNet-50 trained on CIFAR-100.

Figure 4 caption -- “(...) requiring for restoring (almost) full generalization in the noise model” -- What “almost” in this case means? These are the details the need precise treatment otherwise the reader has to guess them and could be easily mislead.



I’m also a bit confused about the discrepancy between the differences of the trends presented in Figure 2 (ResNet34 trained on CIFAR-10 and CIFAR-100). The architecture is the same, however, the trends are radically different. Could you please elaborate on the source of these differences? Did you run any further analysis trying to understand it?

Figure 3 -- captions of subfigures do not include the datasets.


Section 5.3.3. -- fix the reference to the Figure/Section at the end of the page.


Could you please elaborate a bit on why would you apply different strategy (e.g. LogitLens) in case of Transformers? Do you see any equivalence of stitching method and LogitLens or is it justified in any other way?

---

### Review · Reviewer_AK2u · 2025-12-18

**Summary Of Contributions:**

This paper presents an excellent work explaining the well-known puzzling behavior of deep neural networks: they can memorize incorrect labels in the training data but still perform well on new test data.
The authors use model stitching as an innovative way to identify where memorization occurs inside a neural network.

Strengths

1. I believe this is the first study to use model stitching to compare each layer in a clean model and a noisy model. It's a practical way to test layer-by-layer functional similarity.

2. The authors provides solid testing results. They tested their method on several different models (CNN, ResNet-18/34/50) and datasets (SVHN, CIFAR-10/100), and it showed the same pattern each time.

3. The authors report the following very important findings based on their solid and innovative experimental design:

a. The authors show that memorization is not limited to the deep layers but Early Layers also learn noise. When the noisy early layers is replaced by the clean layers, the model often stopped recalling the wrong random labels. This provides evidence that fixing the bottom of the network is an effective way to prevent noise from propagating through the model.

b. The authors also show the importance of fixing the bottom layers as small errors/noise in the early layers get bigger and bigger as they move up.

Weaknesses

1. The experiments only check vision models (CNNs and ResNets) for classification tasks. They do not test text-based models or Transformers. But LLMs and Transformers are very popular today.

2. The experiments assume already have a perfect clean model to swap layers. But a perfect noise-free model is probably not available in real-world scenarios.

3. In Appendix A.1, the authors admit that swapping layers is difficult in the real world because rarely have a perfect clean model as a replacement. Therefore they tried to fix the noise by retraining the specific layers instead of stitching in new ones. But they find that retraining the specific layers is hard to do well compared to swapping it out as authors mentioned "answering this question needs a method to fix layer-wise noise in the model, which itself is challenging and beyond the scope of our work." and "However,
we do not observe more than 80% accuracy on memorized inputs, showing limitations of fine-tuning
in correcting underlying noise." So overall, the paper successfully identifies the root cause of memorization, but it does not provide a practical solution for fixing it without a perfect clean reference model.

**Audience:**

Yes

**Audience Explanation:**

The findings of this paper would be of interest to the TMLR audience focused on deep learning, interpretability, and robust learning.

**Claims And Evidence:**

Yes

**Claims Explanation:**

The authors provides solid testing results. They tested their method on several different models (CNN, ResNet-18/34/50) and datasets (SVHN, CIFAR-10/100), and it showed the same pattern each time.

**Requested Changes:**

The paper already provides solid analysis and evidence regarding the layer-wise dynamics of memorization. It's better for the authors to address and explain more on how to resolve the limitations mentioned in the Weaknesses 1., 2., 3.

---

### Review · Reviewer_vyRE · 2025-12-21

**Summary Of Contributions:**

The paper studies the extent of memorization (mem) and generalization (gen) in CNNs for image classification tasks with datasets like CIFAR and SVHN. Specifically, it studies the effect of different kinds of stitched models on the mem and gen performance, where the stitching happens between a noisy base and a clean head or vice-versa. The paper conducts several experiments to study the effect of the depth of the base on the mem and gen performance in different CNN arcitectures--with and without skip connections, and reports the observed trends. The trends appear to be consistent between the degree of generalization and that of memorization, i.e., configurations that lead to better gen lead to worse mem and vice-versa. Furthermore, the experiments indicate that the extent of gen and mem for the same depth depends on the architecture and the dataset.

**Audience:**

Yes

**Audience Explanation:**

The paper provides useful analysis on how the degree of memorization (mem) and generalization (gen) are related to the layer count of the base and the head in stitched models with CNN architecture and simple image classification tasks with datasets like CIFAR and SVHN. This study could be valuable to the community for better understanding the mem and gen capabilities of CNNs for image classification.

**Broader Impact Concerns:**

Please see para 1 in "Requested Changes".

**Claims And Evidence:**

Yes

**Claims Explanation:**

The paper conducts rigorous eperiments in support of its claims. Specifically, to test both memorization (mem) and generalization (gen) it experiments with different architectures and datasets. The observed trends seem to be consistent across datasets and architectures. This makes the observations well-justified.

**Requested Changes:**

Given the prevalence of transformer-based models in modern computer vision / AI and more complex tasks of interest, like question answering, generation, low-level vision, 3D vision, video understandinging, etc., I am seriously concerned about the usefulness of this work. The paper can be made stronger and more relevant by adding experiments and extending the study to cover the following aspects:
1. other architectures like transformers, state space models
2. more complex tasks like visual question answering, image/video generation, 3D scene reconstruction / novel view synthesis, object detection and segmentation, video understanding and captioning, or other tasks that are currently of interest in the vision community

I am also curious how the study would extend to image classification with large-scale datasets like Imagenet.

---

### Decision · Action_Editor_SBzB · 2026-01-20

**Recommendation:** Accept as is

**Audience:**

Yes

**Audience Explanation:**

I believe the main conclusions are interesting and demonstrated on a traditionally accepted scope of datasets and architectures to be trusted and potentially expanded upon.

**Claims And Evidence:**

Yes

**Claims Explanation:**

The paper explores how memorization impacts the representations across layers. The main novelty is applying stiching: inserting linear layers that transform the representation of the source model and are trained to minimize the loss of the stiched model when using the source layer representation.

A key strength of the paper are the clear findings that are proven in a robust way. In particular, Reviewers have found interesting that early layers when “cleaned” (i.e. replaced through the stiching process with clean layers) leads to significant restoration of performance, despite keeping the later layers.

A commonly voiced concern was that experiments were limited in scope in terms of architectures and tasks (i.e. only image classification). In response, Authors have extended the scope to include the Transformer architecture. The Authors argued that extending to additional tasks is beyond the scope of the paper. This was a key issue for the negative reviewer vyRE.

Another issue is that the paper is not prescriptive. The paper introduces an interesting diagnostic method, makes seemingly valuable observations, but it is less clear what are the implications. Finally, it was raised by AVA2 that the stiching approach used in the paper was not compared to other CCA/CKA, which are more popular. These issues were not fully addressed during the discussion phase.

All in all, it is a borderline work, and I agree with the issues raised by Reviewers. Having said that, I believe the findings, even if potentially limited in scope, are correct and interesting. As such, I am happy to recommend acceptance of the work.